# SAEdit: Token-level control for continuous image editing via Sparse AutoEncoder

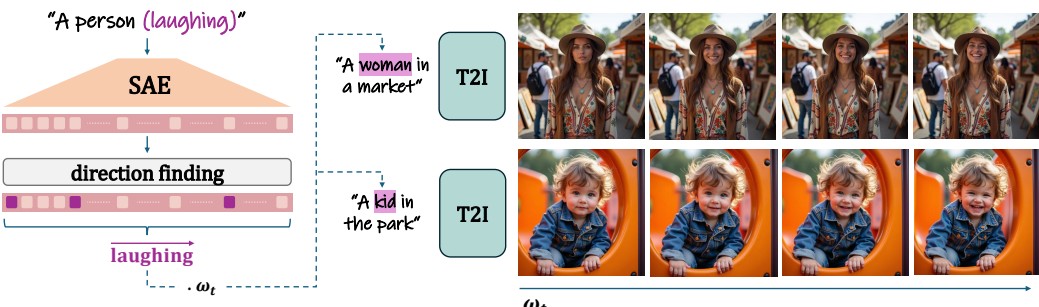

Figure 1: We train a Sparse AutoEncoder (SAE) to lift the text embeddings into a higher-dimensional space, where we identify disentangled semantic directions (e.g. for laughing). These directions can then be applied to specific tokens within the input of a text-to-image model to facilitate continuous image editing. As shown on the right, our token-level editing steers the model to incorporate the relevant attribute (laughing) into the subject in the image that corresponds to the chosen token (e.g., "woman" or "kid"), while allowing the attribute's intensity to be continuously adjusted through a scale factor, $\omega_t$.

## ABSTRACT

Large-scale text-to-image diffusion models have become the backbone of modern image editing, yet text prompts alone do not offer adequate control over the editing process. Two properties are especially desirable: disentanglement, where changing one attribute does not unintentionally alter others, and continuous control, where the strength of an edit can be smoothly adjusted. We introduce a method for disentangled and continuous editing through token-level manipulation of text embeddings. The edits are applied by manipulating the embeddings along carefully chosen directions, which control the strength of the target attribute. To identify such directions, we employ a Sparse Autoencoder (SAE), whose sparse latent space exposes semantically isolated dimensions. Our method operates directly on text embeddings without modifying the diffusion process, making it model agnostic and broadly applicable to various image synthesis backbones. Experiments show that it enables intuitive and efficient manipulations with continuous control across diverse attributes and domains.

## 1 INTRODUCTION

Large-scale text-to-image diffusion models have revolutionized the field of image synthesis (Ramesh et al., 2022; Rombach et al., 2022; Saharia et al., 2022). Consequently, they have become a powerful foundation for a wide array of image manipulation and editing methods (Meng et al., 2022; Hertz et al., 2022; Tumanyan et al., 2023; Cao et al., 2023). These methods have demonstrated remarkable success in a range of edits, including adding new elements, replacing parts of the scene, and modifying the attributes of existing objects. Two properties are particularly desirable in such edits: disentanglement, which ensures that modifying one attribute does not unintentionally affect others, and continuous control, which allows adjusting the magnitude of the edit.

While there has been significant progress in achieving disentangled editing, finding controllable representations that enable edits which are both disentangled and continuous remains a major challenge. Text prompts alone struggle to provide this level of control, as their discrete nature prevents

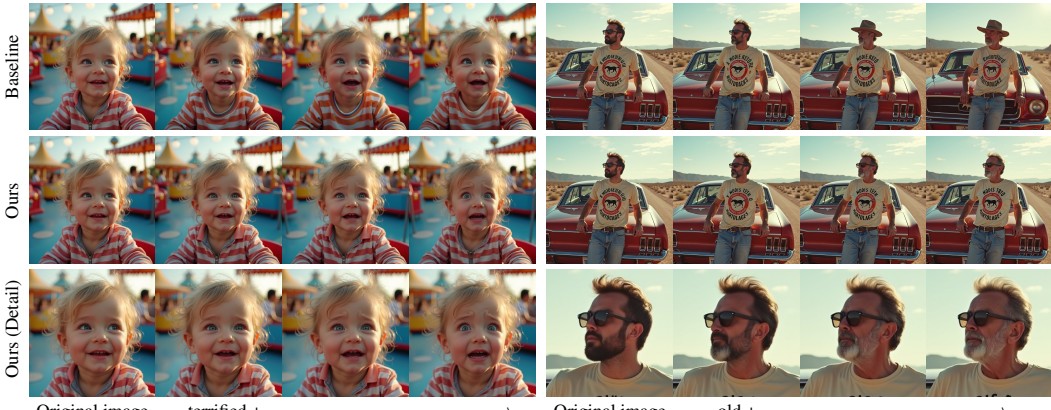

Figure 2: Naïvely applying T5 edit direction (top) by interpolating T5 embedding of target edit, introduces entangled changes that may distort the scene. This can appear as an insufficient edit (left example) or as the modification of unwanted elements (right example). In contrast, edit directions found by the SAE (bottom) yield disentangled edits that preserve identity and achieve the intended modification.

smooth intensity adjustment and their holistic influence often leads to unintended changes. For example, to control the intensity of a smile, a user must resort to distinct coarse categorical descriptions like "a slight smile" versus "a wide grin", rather than smoothly varying the intensity. This limitation motivates research into underlying semantic control mechanisms that are both continuous and disentangled.

In pursuing this goal, some works have focused on general, training-free methods that manipulate the diffusion model's internal representations (Dalva et al., 2024; Guerrero-Viu et al., 2024; Baumann et al., 2025). While versatile, these techniques often struggle with disentanglement, where an edit intended to be local inadvertently causes widespread, undesirable changes to the overall image style and composition. To achieve higher fidelity, other approaches have pursued task-specific optimization, training a dedicated module for each edit (Gandikota et al., 2023; Sridhar & Vasconcelos, 2024; Dravid et al., 2024), with the module's weights acting as the controllable representation for the edit. However, while often producing high-quality results, this strategy is inherently unscalable, demanding a unique training pipeline for every possible modification.

In this work, we propose a method for disentangled and continuous image editing through the fine-grained manipulation of text embeddings at the token-level. Our approach leverages a Sparse Autoencoder (SAE) (Cunningham et al., 2023), an unsupervised model trained to reconstruct its input from a sparse, high-dimensional latent space. The sparsity of this latent space induces semantically disentangled dimensions, which in turn enable the discovery of meaningful editing directions for each token.

Specifically, we derive an edit direction in the SAE's space by comparing the sparse representations of two prompts that differ by the desired edit description (e.g., "a person" and "a smiling person"), identifying the entries most correlated with the change. We then construct an edit-specific direction as a sparse vector that modifies only these highly relevant entries.

This disentangled direction is added to the sparse representation of the prompt, and can be scaled to continuously control the magnitude of the target attribute, while preserving the rest of the image. This approach leverages the SAE to uncover disentangled directions that are difficult to identify directly in the raw embedding space, as qualitatively demonstrated in Figure 2.

Our method operates solely on the text embeddings, leaving the denoising process untouched. In this setup, the diffusion model serves merely as a renderer: it receives the edited semantic instructions and translates them into a visual output. As a result, the method is model-agnostic and can be applied to any text-to-image backbone that shares the same text encoder, without additional training or fine-tuning.

Through extensive experiments, we demonstrate the effectiveness of our method in providing both continuous and highly disentangled semantic edits. We validate the versatility of our approach by ap-

plying the same framework to various generative models, including two image synthesis backbones, without any model-specific training. Importantly, we show that our method enables a wide range of intuitive, magnitude-controlled manipulations from simple text commands, as demonstrated in Figure 1. We further show that our method can be applied to real images using inversion techniques.

## 2 RELATED WORK

**Image Editing with Diffusion Models**   The success of diffusion models in image synthesis (Ho et al., 2020; Song et al., 2022; Ramesh et al., 2022; Saharia et al., 2022; Song et al., 2021; Black Forest Labs, 2024; Podell et al., 2023) has led to their widespread adoption for the more challenging task of real image editing. Unlike pure generation, editing requires a careful balance between preserving an image's original attributes and introducing controlled, text-guided changes. Common strategies include manipulating the denoising process through feature injection (Hertz et al., 2022; Parmar et al., 2023; Tumanyan et al., 2023; Cao et al., 2023; Alaluf et al., 2023; Patashnik et al., 2023) or applying partial noise schedules with a new text condition (Meng et al., 2022; Huberman-Spiegelglas et al., 2023; Tsaban & Passos, 2023; Brack et al., 2023b; Deutch et al., 2024; Rout et al., 2024). A key requirement for applying these methods to real images is an inversion technique that can find an initial noise capable of reconstructing the image (Dhariwal & Nichol, 2021; Mokady et al., 2022; Miyake et al., 2023; Han et al., 2024; Garibi et al., 2024; Samuel et al., 2024; Jiao et al., 2025; Kadosh et al., 2025).

**Continuous Image Editing with Diffusion Models**   A challenge in this area is achieving fine-grained, continuous control over semantic attributes. To achieve this kind of control some methods perform Task-specific optimization methods, which yield high-fidelity, disentangled edits but are not scalable, requiring a separate, costly process for each new attribute, such as training a dedicated LoRA adapter (Gandikota et al., 2023), optimizing a text token (Sridhar & Vasconcelos, 2024) or to train numerous person-specific DreamBooth LoRAs (Ruiz et al., 2023) and then trains a classifier in the latent space (Chang et al., 2025) or in the weights' space (Dravid et al., 2024). Conversely, training-free methods that discover semantic directions in existing latent spaces (Dalva et al., 2024; Guerrero-Viu et al., 2024; Baumann et al., 2025; Garibi et al., 2025; Dorfman et al., 2025; Brack et al., 2023a) are general-purpose but often struggle with the precision and disentanglement of specialized models. Other works (Gandikota et al., 2025; Dalva & Yanardag, 2023) explore unsupervised discovery of a model's latent variations but are not designed for direct, text-guided editing. Our work aims to bridge this gap, offering a general framework that provides the disentangled control of task-specific methods without the need for per-edit training.

## 3 PRELIMINARY - SPARSE AUTOENCODERS

Sparse Autoencoders (SAEs) are neural architectures designed to learn interpretable and disentangled high-dimensional latent representations (Cunningham et al., 2023). An SAE typically consists of a simple encoder, often a single linear layer with a non-negative activation, and a linear decoder. The model is trained with a dual objective:

$$\mathcal{L} = \mathcal{L}_{\text{rec}} + \alpha \cdot \mathcal{L}_{\text{sparse}}, \tag{1}$$

where $\mathcal{L}_{\text{rec}}$ is a standard reconstruction loss, and $\mathcal{L}_{\text{sparse}}$ is a set of regularization terms that encourages the latent representation to be sparse. This sparsity constraint encourages the SAE to learn a dictionary-like representation, where a small set of active latent features often corresponds to a distinct semantic attribute of the input. This property makes SAEs a powerful tool for interpreting the otherwise dense and opaque hidden states of large language models.

Consequently, SAEs have been successfully applied to the internal states of large language models to uncover meaningful, semantic features (Bricken et al., 2023; Gao et al., 2024; Cunningham et al., 2023). For example, Bricken et al. (2023) found that certain features in the sparse representation are active only when specific entities, such as "US presidents," are mentioned in the text. Identifying which features correspond to specific concepts enables model steering, allowing for direct control over model behavior by manipulating its internal activations (Arad et al., 2025; Bayat et al., 2025). The basic SAE framework can be extended with more advanced variants and sparsity regularization techniques, which are detailed further in Section C.

Recent works have integrated Sparse Autoencoders (SAEs) into diffusion models for various distinct purposes. For instance, *One-Step is Enough* (Surkov et al., 2025) trains SAEs on SDXL cross-

attention layers to extract interpretable features from the model's internal representations. Furthermore, *Concept-Steerers* (Kim & Ghadiyaram, 2025) and *SAEuron* (Cywiński & Deja, 2025) utilize SAEs to steer generation toward safer outputs and enable concept unlearning, respectively. However, these methods do not support controllable, continues image editing.

## 4 METHOD

We present a method for text-driven image editing that provides both disentanglement and continuous control. Our approach is based on manipulating the text embeddings of a frozen text-to-image model. We train a Sparse Autoencoder (SAE) on these embeddings, which provides a space in which disentangled directions corresponding to semantic attributes can be found. Editing is then performed by adjusting the embeddings along these directions to achieve controlled manipulations.

Specifically, given a frozen text encoder, we train a Sparse Autoencoder (SAE) on its output embedding space (details in Sec. 4.1). The SAE is composed of an encoder, $\mathcal{S}_{enc}$, and a decoder $\mathcal{S}_{dec}$. The encoder maps dense text embeddings into a high-dimensional, disentangled latent space where distinct semantic concepts are isolated, while the decoder reconstructs the original embedding from this sparse representation. Once trained, manipulations are applied directly in this sparse SAE's space by adjusting specific entries in the latent representation. The modified representation is then passed through the SAE's decoder to recover an edited text embedding, which can be fed into any compatible text-to-image model (e.g., Flux) that uses the same text encoder architecture. In this way, the SAE acts as a lightweight, pluggable module that enables disentangled and semantic control over the final generated image.

The editing direction is obtained from a source prompt $\mathcal{P}_{src}$ (e.g. a "man") and target prompt $\mathcal{P}_{tgt}$ (e.g. "a smiling man") , details in Sec. 4.2. We apply the edit direction by multiplying it with a scale factor and adding it to the sparse representation of the specific source token in $\mathcal{P}_{src}$ to be edited (e.g. the "man" token). The magnitude of the edit is dictated by this scale factor, allowing for continuous control over the attribute's intensity (details in Sec. 4.3).

We demonstrate our method on the T5 text encoder (Raffel et al., 2023), which is widely adopted as the text conditioning module in many state-of-the-art text-to-image models. For the image generation backbone, which acts as a renderer for our text embedding manipulations, we primarily use the Flux (Black Forest Labs, 2024) diffusion transformer (DiT).

### 4.1 SAE TRAINING

We train our Sparse Autoencoder (SAE) on a dataset of text embeddings. To create this dataset, we first process a corpus of text prompts through the frozen T5 text encoder and collect the resulting token embeddings, excluding padding tokens. Notably, unlike typical SAE applications that focus on intermediate transformer layers, we train our SAE on the final output of the text encoder, as these are the exact representations that are continuously processed by the Diffusion Transformer (DiT) throughout the denoising steps.

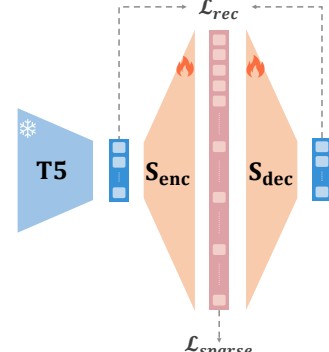

Figure 3: We train the Sparse Autoencoder on token embeddings obtained from a frozen T5 encoder, using reconstruction and sparsity losses.

The SAE is trained on the embeddings of individual text tokens, using the objective function from Eq. 1, the process illustrated in Fig. 3. Here, $\mathcal{L}_{rec}$ is the standard reconstruction loss (e.g., Mean Squared Error) between the SAE's input and output embeddings. We control the target level of sparsity via another hyperparameter which sets the desired number of non-zero activations for each token's latent code.

### 4.2 OBTAINING AN EDIT DIRECTION

Motivated by prior work on SAEs in language models, which shows that specific entries in the sparse representation activate only in the presence of particular semantic attributes (Cunningham et al., 2023; Gao et al., 2024), we aim to detect such entries to construct disentangled directions in the SAE's latent space for image editing. To do so, we use a source prompt $\mathcal{P}_{src}$ (e.g. "a woman") and a target prompt $\mathcal{P}_{tgt}$ (e.g. "a woman laughing"). We first encode all text tokens in both prompts using the SAE encoder, $\mathcal{S}_{enc}$, to obtain sparse token representations. Since it is unknown apriori

(happy) Man sits in bar

Man sits in bar → T5 → $S_{enc}$ → ⊕ → $\omega_t$ → $\overrightarrow{happy}$ → $S_{dec}$ → DiT →

Figure 5: Applying the edit direction. An aggregated edit direction is scaled to adjust edit magnitude and applied to the sparse representation of the relevant source token (e.g., man). The result is then decoded back into the T5 embedding space, and used to condition the text-to-image model.

which tokens hold the semantic information for a concept (Kaplan et al., 2025), we use element-wise max-pooling to aggregate their sparse representations into a single, sparse vector for each prompt. As $\mathcal{P}_{src}$ and $\mathcal{P}_{tgt}$ are semantically similar except for the edited attribute, the activated entries in $\text{maxpool}(S_{enc}(\mathcal{P}_{src}))$ and $\text{maxpool}(S_{enc}(\mathcal{P}_{tgt}))$ should overlap substantially, with their differences centering around entries corresponding to the edit-specific attribute.

To identify the entries that correlate with the requested edit, we compute an entry-wise ratio, $R$, between the source and target prompt:

$$R = \frac{\text{maxpool}(S_{enc}(\mathcal{P}_{tgt}))}{\text{maxpool}(S_{enc}(\mathcal{P}_{src})) + \epsilon}, \tag{2}$$

where $\epsilon$ is a small constant added for numerical stability. The entries in $R$ with the highest values correspond most strongly to the edit-specific attribute. Next, to isolate these key entries , we normalize the ratio vector, $R^{norm} = R/\max(R)$, and apply a predefined threshold $\rho \in [0,1]$. This yields a set of indices, $M$, corresponding to the most relevant entries for the edit:

$$M = \{i \mid R_i^{norm} > \rho\}. \tag{3}$$

Finally, we use this set of indices to construct the disentangled edit direction, $d_{edit}$, as a sparse vector, as illustrated in Fig. 4. The direction is defined to be zero everywhere except at the identified indices, where it takes its values from the target representation:

$$[d_{edit}]_i = \begin{cases} [S_{enc}(\mathcal{P}_{tgt})]_i & \text{if } i \in M, \\ 0 & \text{if } i \notin M \end{cases} \tag{4}$$

**Improving direction's robustness** To en-

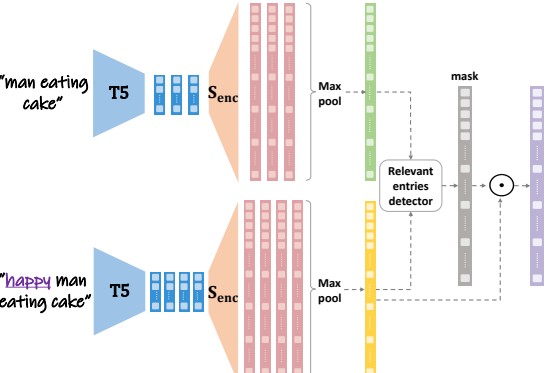

Figure 4: Extracting Edit Directions. We derive an edit direction from a prompt pair that isolates a single attribute. Both prompts are encoded with the SAE, and their token representations are aggregated via max-pooling. By comparing the two resulting sparse vectors, we identify the key features corresponding to the desired change. The final edit direction is a sparse vector composed of only these key features, taken from the target prompt's representation.

hance the robustness of our derived edit directions, we aggregate information from a set of multiple source-target prompt pairs rather than relying on a single pair. Given a desired edit, defined by the pair of texts descriptions $\mathcal{P}_{src}$ and $\mathcal{P}_{tgt}$, we use an LLM to construct $N$ sentence pairs that share the same underlying semantic relationship. This process, generalizes the specific edit into an abstract concept. For example, to create a direction for "happiness", the LLM generates pairs that add this attribute to various contexts, such as ("man on the beach", "happy man on the beach") and ("man eating cake", "happy man eating cake"). We then apply our direction-finding procedure to each of the $N$ prompt pairs, resulting in a set of $N$ steering vectors $\{d_i\}_{i=1}^N$. These vectors are stacked to form a direction matrix: $D = [d_1, \ldots, d_N]^T$. To extract the most prominent and consistent direction representing the shared attribute across all examples, we perform Singular Value Decomposition (SVD) on $D$. The singular vector corresponding to the largest singular value is then selected as our final, robust edit direction $d_{\text{edit}}$.

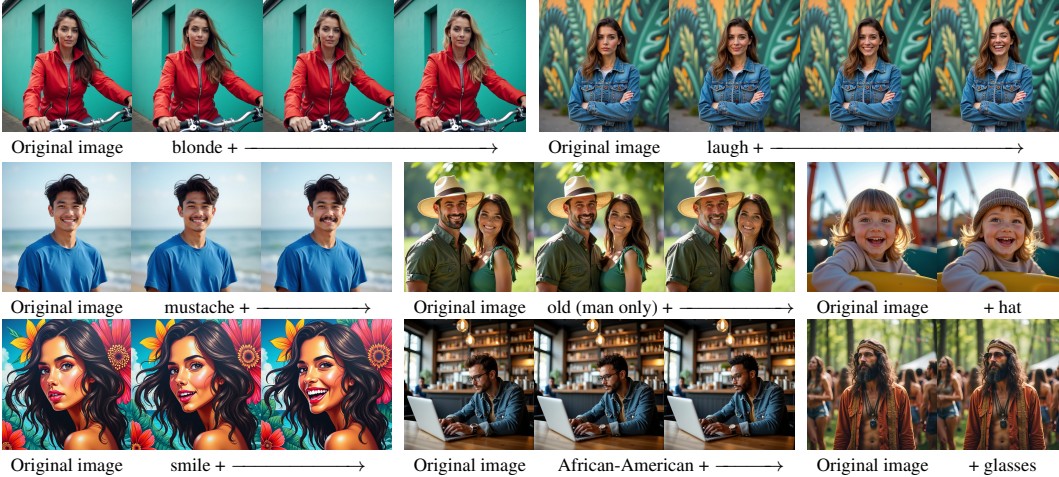

Figure 6: Qualitative Results. Our method enables a diverse range of continuous and disentangled semantic edits across various image styles. We demonstrate the ability to add attributes (e.g., mustache, glasses), change expressions (smile, laugh), and perform highly localized edits, such as modifying the age of only one person in a scene.

### 4.3 APPLYING THE EDIT DIRECTION

Once the edit direction $d_{edit}$ is derived, we apply it to the source prompt $\mathcal{P}_{src}$. To ensure the manipulation is localized, we modify only the embedding of the specific token to be edited (e.g., the "woman" token), which we denote as $e_{tgt}$. The magnitude of the edit is controlled by a scalar factor $\omega$, allowing for continuous, fine-grained control over the attribute's intensity.

The final, edited text embedding for the token, $e'_{tgt}$, is produced by first encoding the original token's embedding with $\mathcal{S}_{enc}$, adding the scaled direction in the sparse latent space, and then decoding the result with $\mathcal{S}_{dec}$:

$$e'_{token} = \mathcal{S}_{dec}(\mathcal{S}_{enc}(e_{tgt}) + \omega \cdot d_{edit}). \tag{5}$$

Setting $\omega = 0$ recovers the original embedding, while progressively increasing $\omega$ strengthens the visual effect. This new token embedding, $e'_{tgt}$, replaces the original in the prompt.

Finally, the manipulated text embeddings are used to condition the renderer. Specifically, for diffusion models, we follow the standard editing approach to preserve the overall structure of the source image. This involves using the same initial noise, $x_T$, that was used to generate the source image, and only substituting the original token embeddings with our modified ones. This ensures that the changes in the final generated image are driven exclusively by our disentangled edit. Fig. 5 provides a schematic of this entire editing pipeline.

### 4.4 INJECTION SCHEDULE

The denoising process in diffusion models operates hierarchically: early timesteps are crucial for establishing the global structure and layout of an image, while later steps refine fine-grained details and textures (Patashnik et al., 2023; Balaji et al., 2023; Cao et al., 2025; Huberman et al., 2025; Yehezkel et al., 2025). Consequently, for fine-grained edits that aim to preserve the original structure, prior work has shown that it is often optimal to begin the editing manipulation only at later timesteps, after the core layout is formed (Huberman-Spiegelglas et al., 2023; Jiao et al., 2025).

Building on this insight, we introduce an exponential injection schedule that applies the edit direction with increasing intensity over time. For a base scale factor $\omega$ and diffusion step $t$, we define the time-dependent scale $\omega_t$ as:

$$\omega_t = \min \left( e^{t \cdot \omega} - 1, \tau \right), \tag{6}$$

where $\tau \in \mathbb{R}$ is a hyperparameter that acts as an upper bound on the edit strength. This exponential formulation offers a key advantage over linear schedules: it applies the edit very gently in the early, structure-defining timesteps and progressively increases its influence as the process moves into the later, detail-refining stages. This gradual application better aligns with the hierarchical nature of image synthesis, preserving global structure while enabling powerful, fine-grained modifications.

## 5 EXPERIMENTS

We conduct extensive experiments to evaluate our method's ability to provide continuous control and disentangled edits that preserve the subject's identity. Similar to prior work, we focus our evaluation

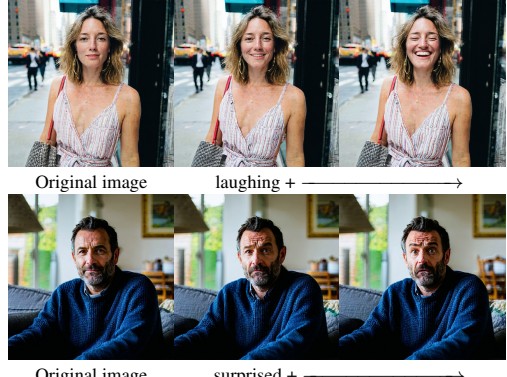

Original image     laughing + ——————————→

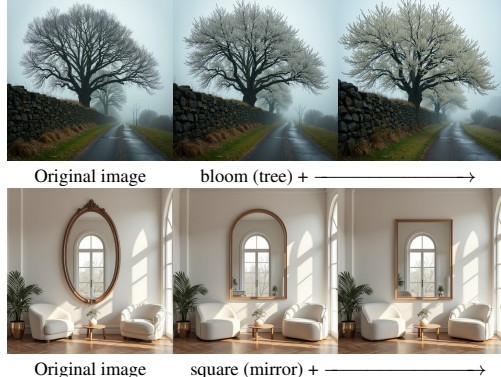

Original image     bloom (tree) + ——————————→

Original image     surprised + ——————————→

Original image     square (mirror) + ——————————→

Figure 7: Results with SD3.5. These demonstrate that our method integrates seamlessly with models relying on T5, enabling consistent and faithful edits across architectures.

Figure 8: Our method's versatility extends beyond human subjects, enabling continuous and disentangled control over object attributes like seasonal appearance and object shape.

on human subjects, a challenging domain that demands strong disentanglement to preserve identity and offers the most meaningful application of continuous magnitude control. To demonstrate its model-agnostic nature, we apply our approach to both Flux (Black Forest Labs, 2024) and Stable Diffusion 3.5 (Esser et al., 2024). Unless otherwise specified, all results are generated using Flux. We also show its applicability to real image editing through integration with standard inversion techniques. For quantitative evaluation, we measure preservation with LPIPS (Zhang et al., 2018) and semantic accuracy with a VQA-Score (Lin et al., 2024). Implementation details in the Appendix A.

## 5.1 QUALITATIVE RESULTS

We present qualitative results generated by our method, SAEdit, in Figures 1, 6, 7 and 8. Figure 6 shows a wide variety of continuous edits on human subjects. Our method successfully changes expressions (e.g., adding a smile), modify attributes (e.g., making hair blonde), and add accessories (e.g., hats or glasses). Crucially, these edits are highly localized. For instance, we demonstrate the ability to modify the age of a single person in a multi-subject image while leaving the other person and the background entirely untouched.

The results also highlight the continuous nature of our control. As shown in the examples, attributes such as the intensity of a laugh or the degree of age can be smoothly scaled. This allows users to precisely tune the magnitude of the desired effect while the rest of the image content is faithfully preserved.

The approach is not limited to human subjects and generalizes to a broad range of semantic concepts, as shown in Figure 8. Finally, to demonstrate the model-agnostic nature of SAEdit, Figure 7 shows that the same edit directions produce consistent, high-quality results when applied to a different T5-based model, Stable Diffusion 3.5. We provide additional qualitative results in Appendix B.1, including examples of our method applied to real images in Appendix B.2 and used for gradual change in style in Figure 26.

## 5.2 ABLATION

Figure 9 provides a qualitative ablation study of our method's components, demonstrating their respective contributions to the final result. As a baseline, deriving an edit direction from a single prompt pair (top row) preserves the subject's identity, but the intended semantic change to the expression is weak and insufficient. Aggregating the direction from $N$ prompt pairs (middle row)

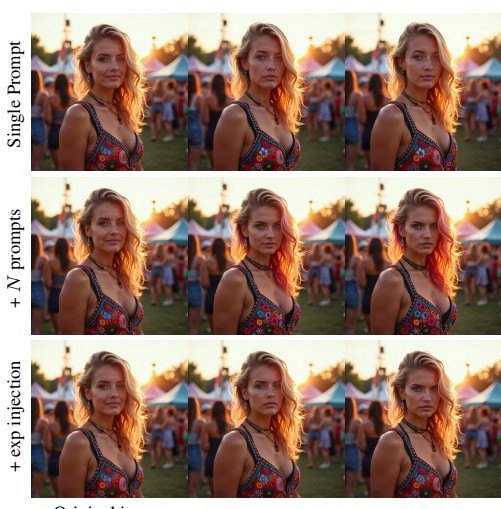

Single Prompt

+ $N$ prompts

+ exp injection

Original image     angry + ——————————→

Figure 9: Ablation study. We demonstrate how each component progressively improves the quality of an 'angry' edit. A direction from a single prompt pair results in a weak edit with unintended modifications. Aggregating N prompts produces a more robust and semantically accurate direction, but can still alter fine details. Adding our exponential injection schedule preserves the original image's details (e.g., the necklace and hair color), yielding the most faithful and disentangled result.

successfully strengthens the edit as required, but causing minor unwanted changes to the hair color, the necklace, and the dress texture. Finally, incorporating our exponential injection schedule (bottom row) resolves this issue by preserving these fine-grained details while maintaining the strong semantic edit, thus achieving a high-quality and disentangled result. Our quantitative ablation study is detailed in Appendix B.4.

## 5.3 COMPARISONS

We evaluate our method against several state-of-the-art approaches for continuous image editing, highlighting its ability to provide disentangled control without per-edit optimization. We compare against methods from both optimization-based and training-free categories. From the optimization-based group, we evaluate Concept Sliders (Gandikota et al., 2023) by using their official SDXL-trained LoRAs as well as LoRAs we trained on the Flux architecture for a direct comparison. We additionally include Prompt Sliders (Sridhar & Vasconcelos, 2024), a textual-inversion (Gal et al., 2022) based method that learns concept embeddings used for adjustable control. In the training-free category, we compare against FluxSpace (Dalva et al., 2024), adjusting its $\lambda_{\text{fine}}$ parameter to control edit magnitude. We also evaluate against Attribute Control (Baumann et al., 2025), a method that proposes both a training-free and an optimization-based variant. For Flux Kontext (Labs et al., 2025), which does not natively support continuous edit scaling, we implement two proxy baselines for magnitude control over the edit strength: the first involves varying the Classifier-Free Guidance (CFG) strength, while the second uses an LLM to generate prompts corresponding to 'light' and 'extreme' versions of each edit (instruction prompts and more details provided in Appendix B.5).

**Quantitative Comparisons** To evaluate the fine-grained and continuous control of our method, we constructed a custom evaluation set. This set is based on 63 images, each generated from a unique prompt created by a large language model (OpenAI, 2025). Each prompt describes a scene containing a person. For each source image, we applied a set of 6 to 8 different semantic edit directions, resulting in 432 unique edit scenarios. To assess the continuity of these edits, we then generated each scenario at 3-5 distinct magnitude levels, producing a final evaluation set of at least 1,296 images per method. The complete list of prompts and edit directions is provided in Appendix B.3. We quantitatively assess our method along three

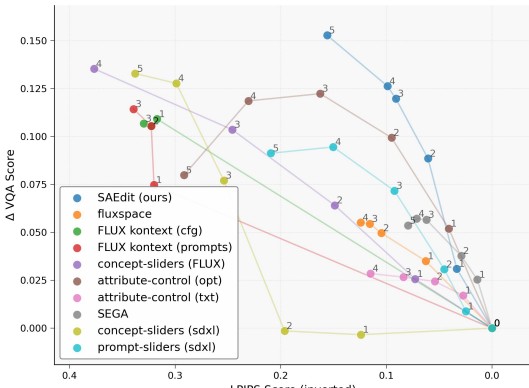

Figure 10: Quantitative comparison. We compare our method to other baselines on image preservation and prompt fidelity (top-right is better).

axes: image preservation, prompt adherence and identity preservation. To measure the image preservation of original content, we use LPIPS (Zhang et al., 2018). To measure prompt adherence with the edit, we compute a VQA-based score (Lin et al., 2024). This score is the delta between the VQA score of the edited image against the target prompt and that of the source image against the same prompt, which isolates the semantic change introduced by the edit. Finally, to evaluate identity preservation, we employ ArcFace (Deng et al., 2022) to calculate the cosine similarity between the subject in the source image and the edited output.

Figure 10 and 20 presents the quantitative comparison between our method and other methods at varying levels of edit intensity. The results demonstrate that our method outperforms all other approaches.

Notably, our zero-shot method is superior even to task-specific techniques that are explicitly trained for each edit type. This indicates that our approach successfully achieves the dual goals of high semantic accuracy for the required edit and strong preservation of the original content. Furthermore, the metrics show a smooth and predictable progression as the edit magnitude increases, confirming that our method provides true continuous control and allows users to precisely tune the intensity of an effect.

Table 1: User Study. Pairwise win rate of our method against other methods.

| Opponent Method | Image Pres. | Prompt Adher. | **Overall** |
|---|---|---|---|
| Flux Kontext (CFG) | 73% | 71% | 70% |
| Flux Kontext (LLM) | 60% | 68% | 70% |
| ConceptSlider (Flux) | 71% | 67% | 71% |
| Flux Space | 59% | 92% | 93% |

**User Study** To complement our quantitative analysis, we conducted a user study to evaluate the perceptual quality of our method against competing approaches. For fairness, we limited our comparison to methods that also use the Flux model, ensuring the source images were as similar as possible. In a pairwise comparison, we presented participants with results from our method and a competing method, showing three distinct levels of edit intensity for each to assess continuous control. Users were asked to state their preference based on three criteria: Image Preservation, Prompt Alignment (which included the gradualness of the effect), and Overall Quality. In total, our user study gathered 390 pairwise comparison responses. More details in Appendix B.7

The results, summarized in Table 1, show that our method was significantly preferred over all other approaches in all categories. This suggests that users found our edits achieve a better balance of successfully applying the desired change while faithfully preserving the original image content.

**Qualitative Comparisons** Figure 12 presents a qualitative comparison between our method and other approaches. While the results for most methods are taken directly from our quantitative evaluation set, we manually optimized the prompts for the Flux Kontext baselines to ensure the strongest possible comparison, as their default outputs were often suboptimal (see Appendix B.6). For example, for the CFG-based baseline, we found the prompt "Make the man look slightly like a kid" with CFG scales of 1.5 and 1.6 yielded the most plausible results. While most of the compared methods operate on the Flux model, Attribute Control and Prompt Sliders use SDXL-based pipelines. For Prompt-Sliders we implemented an inversion procedure to enable real-image editing within their framework, and for Attribute-Control we use their official real-image editing pipeline based on SDXL Turbo. In both cases, we present the reconstructed inversion output as the "original image" for a fair comparison across methods. Additional qualitative comparisons with Attribute Control and Prompt Sliders are provided in Figure 25. In this configuration, inversion is applied only by our method, while the SDXL-based baselines generate their results directly. The visual comparison highlights the superior disentanglement of our method. For instance, in contrast to Concept-Sliders, our approach achieves a perfect reconstruction of the subject's jacket while applying the desired edit. Similarly, when compared to Flux Kontext, our method successfully modifies the subject's age in a more natural and gradual manner, demonstrating more precise control over the semantic attributes. More results in Appendix B.6.

## 5.4 Integration With Flux-Kontext

Since our method operates directly on the semantic text-embedding space, it can be seamlessly incorporated into instruction-based models. We demonstrate this capability using Flux-Kontext for gradual real-image editing (Figure 24). The application mechanism, however, differs from the standard text-to-image setting. The direction finding mechanism remains the same, but the target token changes. In the text-to-image case the edit direction is injected into

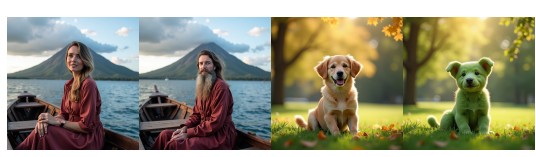

Orignal Image    + beard    Orignal Image    + green

Figure 11: Our method struggles with out-of-distribution (OOD) edits that conflict with strong priors in the base model. For example, applying a "beard" edit changes the woman into a man (left), while making the dog "green" results in an unnatural, animated-style dog (right).

the embedding of the target subject token, whereas instruction-based models require modifying the instruction tokens that encode the editing operation itself.

To enable continuous control in this setting, we apply the edit direction to the tokens corresponding to the action described by the instruction. For example, for an instruction such as "make the man laugh", we subtract the calculated laugh direction from the associated instruction token ("laugh").

This reduces the strength of the directive, providing fine-grained and gradual control over the resulting edit intensity.

### 5.5 LIMITATIONS

While our method identifies robust and disentangled edit directions, we observe that further refinement is sometimes possible. For certain complex edits, manually selecting or de-selecting a few specific entries in the sparse direction vector can yield even more disentangled results.

In addition, our method's ability to disentangle is constrained by the inherent biases of the underlying text-to-image model. When an edit is requested that is strongly out-of-distribution (OOD), our approach can fail to maintain disentanglement. As shown in Figure 11, attempting to add a 'beard' to a "woman" results in the subject's perceived gender being changed to male. Similarly, making a dog "green" alters its texture to appear unnatural and cartoon-like. We hypothesize these failures occur because the SAE cannot fully separate concepts that are fundamentally entangled in the base model's worldview.

### 6 CONCLUSIONS

In this work, we introduced a novel framework that provides both disentangled and continuous control for text-to-image editing. Our method leverages a Sparse Autoencoder (SAE) on text embeddings to create a sparse representation where semantic attributes are isolated. This sparse representation is the key to our method's success. Having isolated individual attributes facilitates disentangled edits, where the subject's core identity is preserved. Our approach enables token-level manipulation, providing fine-grained and continuous control over the magnitude of a given attribute.

A key advantage of our design is that editing is decoupled from rendering: we modify only the text embedding, enabling any compatible text-to-image backbone model to act as the renderer. SAEs are primarily known for their role in interpretability of language models, yet in this work we demonstrate that they can be harnessed for image generation, yielding fine-grained editing capabilities. Image editing has recently seen remarkable progress, yet precise fine-grained control remains an open challenge, and we believe this work will encourage further advances in that direction.

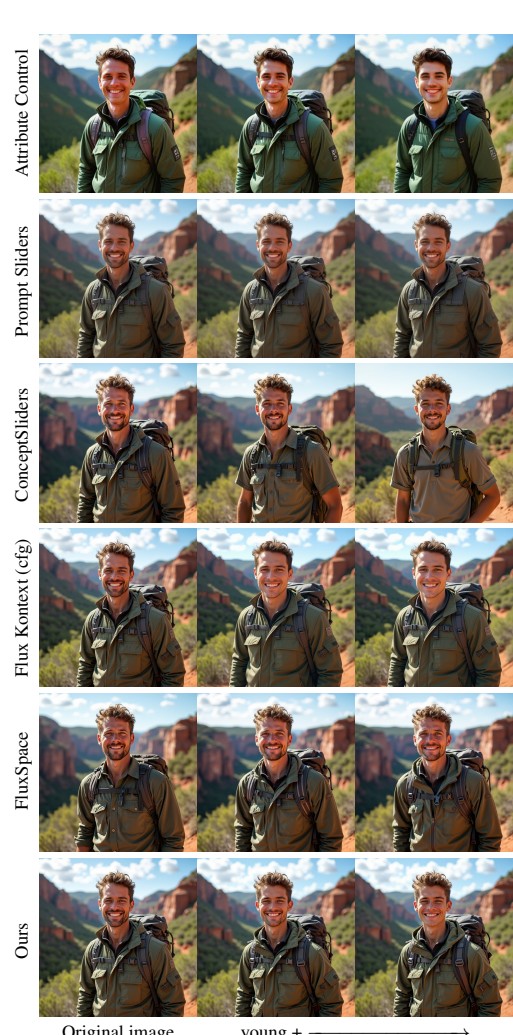

Figure 12: Each row showcases the results of a different editing method for the same edit. Our method (bottom row) produces a more disentangled result that better preserves the subject's identity compared to the competing approaches.

## ETHICS STATEMENT

We acknowledge the ethical considerations inherent in powerful text-to-image models. Our method, like other generative technologies, is a dual-use tool that can be used for both beneficial creative purposes and potential misuse. The underlying text and image models we build upon are trained on large-scale internet data and are known to contain societal biases, which can lead to the generation of stereotypical or harmful content. Furthermore, the ability to manipulate images carries the risk of being used for creating misinformation or other malicious synthetic media.

We have developed this work with the goal of empowering creative applications. We strongly advocate that any deployment of this technology must be accompanied by robust safety filters and content moderation systems to mitigate the risks of generating harmful or biased outputs. We are committed to the principles of responsible AI development and encourage continued research into the safety, fairness, and transparency of generative models.

## REPRODUCIBILITY STATEMENT

To ensure the reproducibility of our results, we will make our code, pre-trained SAE models, and the derived edit directions publicly available upon publication. Our method is built upon the publicly available T5-XXL text encoder, and our experiments use the official model weights for Flux.dev and Stable Diffusion 3.5. Key details for training the Sparse Autoencoder, including all hyperparameters such as the learning rate, sparsity coefficient $\alpha$, and the target number of active features, are provided in Appendix A. The edit directions were derived using 100 prompt pairs generated by GPT for each concept. The complete set of prompts and edits used in our custom evaluation benchmark is included in the supplementary materials to facilitate direct and fair comparisons.

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

# APPENDIX

## A  IMPLEMENTATION DETAILS

We illustrate our method with the T5-XXL text encoder (Raffel et al., 2023), which is utilized by state-of-the-art text-to-image models such as Flux.dev (Black Forest Labs, 2024) and Stable Diffusion 3.5 (Esser et al., 2024). To train the SAE, we compiled a dataset from two sources: the DiffusionDB dataset (Wang et al., 2022), containing 2M general image captions, and the HumanCaption-10M dataset (Dai et al., 2024), which provides 10M captions focused on humans. The combined training set consists of 12M text prompts, totaling approximately 800M text tokens after filtering.

The dimension of the SAE's latent space is set to 65,536, and the target number of active entries for each token is 300. We trained the SAE for 200,000 steps using the Adam optimizer with a learning rate of 0.003. The weight for the sparsity loss, $\alpha$ (from Eq. 1), was set to $\frac{1}{32}$.

For each edit, the corresponding direction was derived using a set of $n = 100$ source and target prompt pairs. These prompt pairs were generated using GPT-5. The parameter $\tau$ (from Eq. 6) used for the exponential injection mechanism was set to be a function of the scale parameter: $\tau = 15 \cdot \omega$.

## B  EXPERIMENTS

### B.1  ADDITIONAL QUALITATIVE RESULTS

Figure 13 showcases the universality of our learned edit directions. We apply the exact same set of four directions (smile, angry, surprised, and old) to four diverse source images, demonstrating that a single direction vector can generalize effectively across different subjects, scenes, and identities.

Figure 14 demonstrates the compositionality of our learned directions, where we independently control a "smile" on the horizontal axis and the addition of "glasses" on the vertical axis. It is evident that these manipulations are highly disentangled, as the subject's identity and all background details remain perfectly consistent across the grid, with only the intended attributes changing. Figure 28 further demonstrates the ability of our method to compose up to 7 edit directions on a single image.

We further demonstrate the compositionality and advanced localization capabilities of our method in Figure 16. The figure showcases the simultaneous application of two distinct edits targeted at different subjects within the same scene. A "laugh" direction is applied to the woman, while an "old" direction is applied to the man. The results across the grid show that each manipulation is successfully confined to its intended target, preserving the background and the non-targeted attributes of each subject without interference.

Figures 23 and 17 present additional qualitative results for continuous editing on human and non-human subjects, respectively.

Figure 27 shows that our method generalizes across seeds, we apply the same edit direction on the prompt "A close up portrait of a man wearing a black coat and a yellow hat, vivid colors, daylight", and we show our results for random seeds 0-3.

### B.2  REAL IMAGE EDITING

Our method's applicability extends to the challenging task of real image editing. To achieve this, we first use a state-of-the-art inversion technique, Uni-Inv Flow (Jiao et al., 2025), to obtain the initial noise corresponding to a given source image. Our SAE-based manipulation is then applied to the text embeddings as previously described. Figure 15 presents several results of this combined approach. As shown, we can apply high-fidelity, continuous edits to real photographs, successfully modifying expressions (cry, laughing) and attributes (old). Importantly, these edits preserve the subject's core identity and background details, demonstrating that our disentangled control is effective even in the demanding context of real image manipulation.

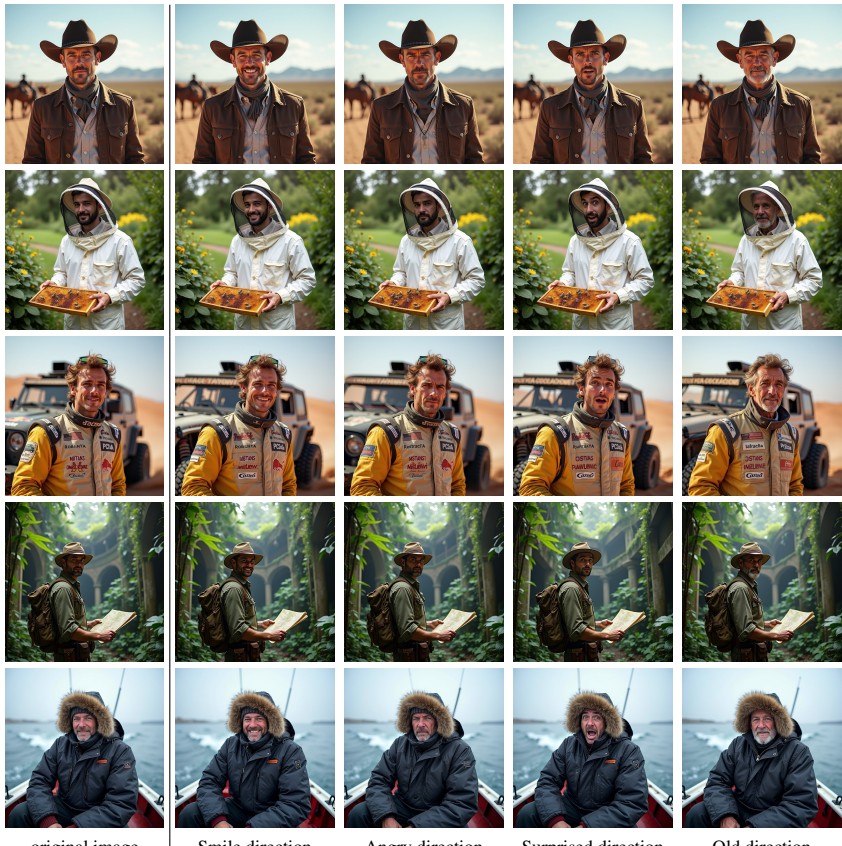

original image | Smile direction | Angry direction | Surprised direction | Old direction

Figure 13: Each row shows a different source image (leftmost column) and its edits along four semantic directions: smile, angry, surprised, and old. The images in each column are generated by adding the same direction, showcasing the generality of the directions found by our method

### B.3 BENCHMARK DETAILS

As mentioned in the main paper, we constructed a custom benchmark for our comparative evaluation. The process began with a large language model (LLM) (OpenAI, 2025), which we used to generate 21 diverse source prompts. For each of these prompts, we generated images using 3 different random seeds, resulting in a set of 63 unique source images. Finally, we applied between 6 to 8 different semantic edits to each source image, depending on the applicability of the edit to the subject. The complete list of source prompts and the specific edits applied to each are detailed in Table 3.

### B.4 QUANTITATIVE ABLATION

To quantitatively measure the contribution of each component of our method, we conduct an ablation study on our benchmark, with results shown in Figure 18. We evaluate three variants of our approach: (1) deriving an edit direction from a single prompt pair, (2) aggregating directions from $N$ prompts but without our proposed injection schedule, and (3) our full method which includes the exponential injection schedule.

The plot of VQA score (prompt alignment) versus LPIPS score (image preservation) reveals the contribution of each component. The single-prompt version serves as our initial baseline and produces a less pronounced semantic change, resulting in a significantly lower VQA score. Aggregating $N$ prompts drastically improves prompt alignment, yielding a much higher VQA score. Our full method, which adds the exponential injection schedule, maintains the high prompt alignment gained from using $N$ prompts while significantly improving image preservation, achieving superior LPIPS scores at all intermediate intensity levels. This validates that both components are crucial for achieving a state-of-the-art balance between edit accuracy and preservation.

.

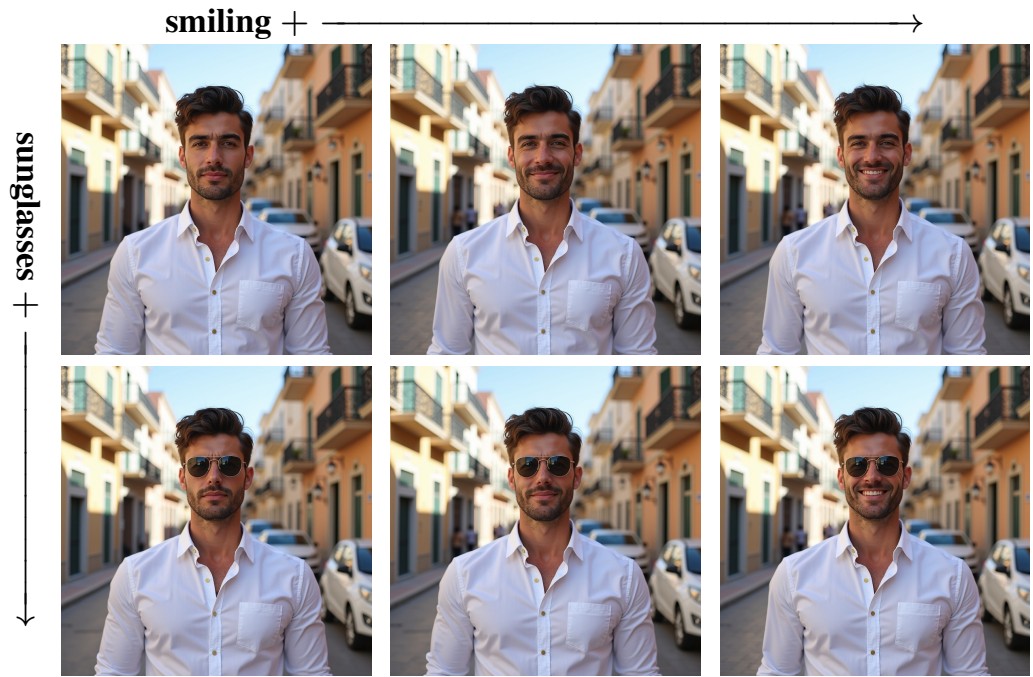

Figure 14: Composing Disentangled Edits. We demonstrates the compositionality of our learned edit directions. Starting from the source image (top-left), we independently control two attributes of the same subject. The horizontal axis continuously controls the "smile" attribute, while the vertical axis adds "glasses". The smooth and accurate results in the grid showcase our method's ability to combine edits.

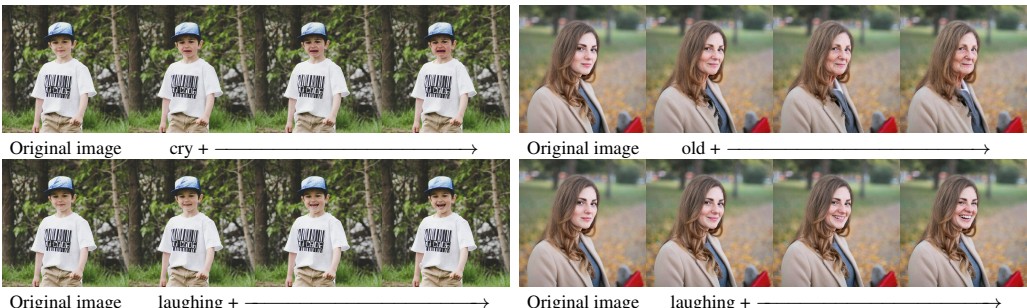

Figure 15: Real Image Editing with Image Inversion. Our method seamlessly integrates with inversion techniques, allowing for high-fidelity edits on real-world images. Leveraging UniFlow (Jiao et al., 2025) to invert the source image into the diffusion model's latent space, we demonstrate continuous control over expressions and attributes. The edits maintain the subject's identity and background fidelity across all intensity levels.

## B.5 FLUX KONTEXT BASELINE

Since Flux Kontext (Labs et al., 2025) lacks a native mechanism for continuous edit scaling, we implemented two distinct proxy baselines to evaluate different edit intensities. The first, which we term Flux Kontext[1] (LLM), controls the edit magnitude by using three different instruction prompts ('light', 'medium', and 'extreme') generated by an LLM, as detailed in Table 2. The second baseline, Flux Kontext[2] (CFG), uses the fixed 'medium' instruction prompt and instead varies the Classifier-Free Guidance (CFG) scale to achieve different levels of edit strength.

**laugh (woman) +** ⟶

**old (man) +**

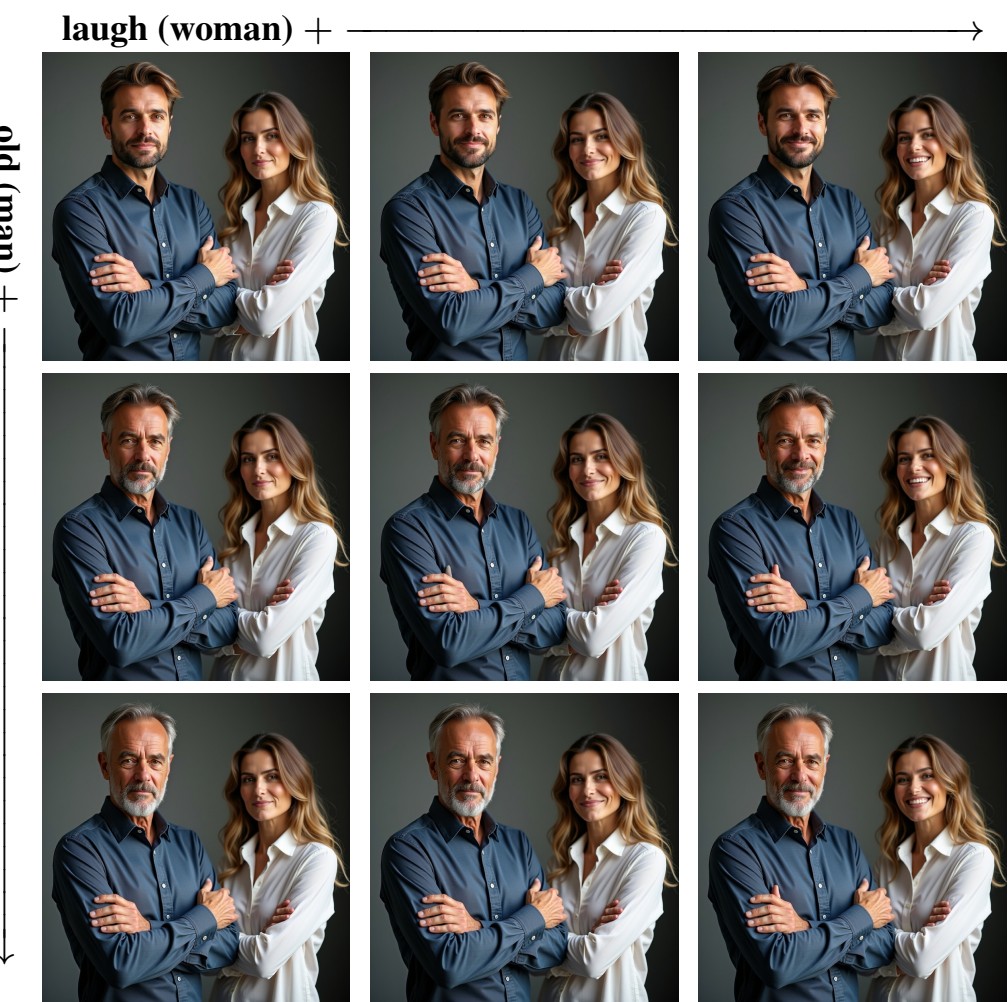

Figure 16: Composition of Edits on Multiple Subjects. We demonstrate our method's ability to apply and compose edits targeted at different subjects within the same image. Starting from the source image (top-left), the horizontal axis applies a "laugh" edit exclusively to the woman, while the vertical axis applies an "old" edit only to the man. The results showcase a high degree of localization and disentanglement, as each edit affects only its intended target without interfering with the other subject or the background.

| Attribute | 1.0 (Low) | 2.0 (Medium) | 3.0 (High) |
|---|---|---|---|
| Bald | make the person balding | make the person bald | make the person completely bald |
| Beard | make the person have short beard | make the person have a beard | make the person have a long thick beard |
| Curly Hair | make the person have slightly curly hair | make the person have curly hair | make the person have very curly hair |
| Laughing | make the person giggle | make the person laugh | make the person laugh hysterically |
| Old | make the person middle-aged | make the person old | make the person very old |
| Smiling | make the person smile slightly | make the person smile | make the person smile broadly |
| Surprised | make the person slightly surprised | make the person surprised | make the person extremely surprised |
| Young | make the person slightly young | make the person young | make the person very young |

Table 2: Textual descriptions of attribute scales used in our comparison with Flux Kontext

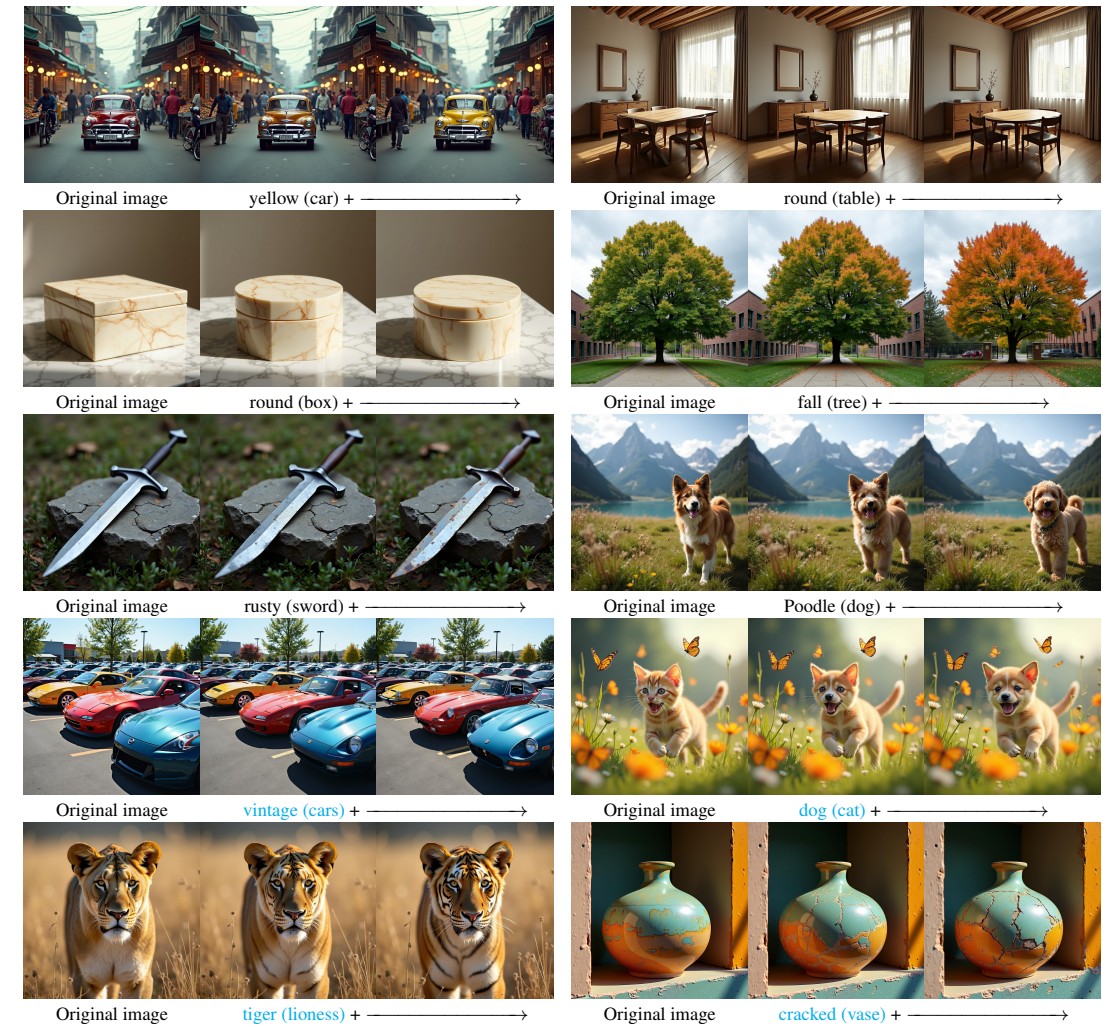

Figure 17: Examples of continuous edits on non-human subjects, showcasing control over seasonal changes, color, and object shape and other attributes.

### B.6 QUALITATIVE COMPARISONS (CONTINUED)

To further evaluate our approach, we provide qualitative comparisons against existing methods, including FluxSpace (Dalva et al., 2024), Concept-Sliders (Gandikota et al., 2023), and two variants of Flux Kontext (Labs et al., 2025): Flux Kontext[1] (LLM), which leverages an LLM to craft prompts for gradual editing, and Flux Kontext[2] (Cfg), which uses the cfg score to guide edits. Results are presented in Figures 21 and 22.

In Figure 21 (left), competing methods fail to introduce a meaningful edit, whereas our method produces a clear and consistent modification. On the right, several baselines either fail to perform the edit or induce significant identity changes. Notably, both Flux Kontext variants are unable to achieve gradual edits and distort subject proportions, often enlarging the head unnaturally. By contrast, our method generates edits that are gradual and identity-preserving.

Figure 22 further illustrates these differences. On the left, competing methods fail to add a beard, produce abrupt transitions, or generate unnatural appearances. Our approach successfully creates a gradual, natural-looking beard. On the right, most baselines again yield non-gradual changes or identity shifts, while our method produces clear, progressive edits that maintain subject identity.

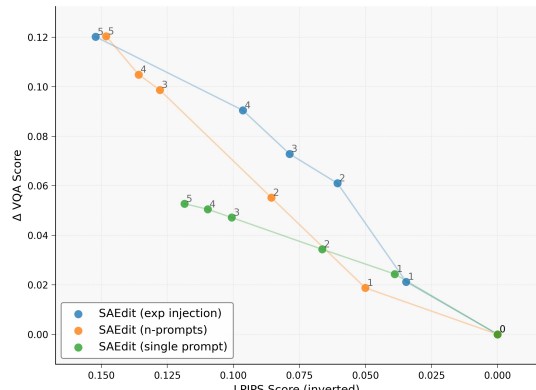

Figure 18: Quantitative Ablation. We compare different versions of our method. (top-right is better).

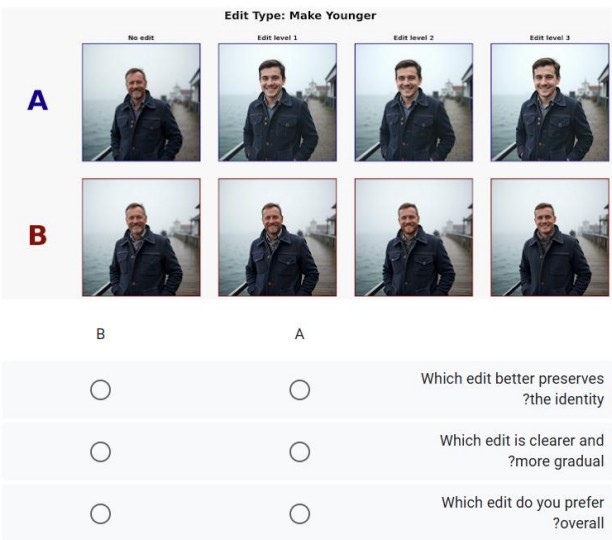

Figure 19: An example of a question in the user study

### B.7 USER STUDY

As reported in the main text, we conducted a user study to further evaluate the perceptual quality of our method. For this study, we randomly sampled 20 edit scenarios from our quantitative evaluation benchmark.

In each question, we performed a pairwise comparison. Participants were shown the three levels of edit intensity from our method alongside the corresponding three levels from a single competing method. They were then asked to choose which set of edits they preferred based on three criteria:

- **Image Preservation:** Which edits better preserves the identity?

- **Prompt Alignment & Graduality:** Which edits is clearer and more gradual?

- **Overall Preference:** Which edits do you prefer overall?

The exact format of the user study interface is shown in Figure 19.

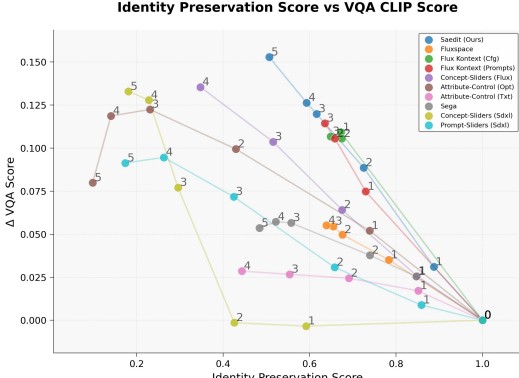

Figure 20: Quantitative comparison. ArcFace is used to compute the identity similarity score between source and target images.

## C  SPARSE AUTOENCODERS - CONTINUE

**Enforcing Sparsity**  Enforcing sparsity in an SAE's latent space is a central challenge that has led to specialized techniques. One prominent method is the BatchTopK operator (Bussmann et al., 2024), a computationally efficient approach that retains only the top $B \times K$ strongest entries across an entire training batch of size $B$. At inference, this operator is replaced by a pre-calibrated global threshold ($\theta$) for consistent behavior on single inputs. A common failure mode with such strong sparsity is the emergence of dead latents, which are entries that cease to activate and in turn degrade the SAE's reconstruction performance. To mitigate this, an auxiliary loss, $\mathcal{L}_{aux}$, can be incorporated (Gao et al., 2024), which encourages these inactive latents to "revive" by tasking them with explaining a portion of the reconstruction error.

**Matryoshka Sparse Autoencoders (MSAEs)**  Bussmann et al. (2025) extend SAEs by learning a single, hierarchical feature dictionary that provides nested representations at multiple levels of granularity. This is achieved by training the model to reconstruct the input using a sequence of nested dictionary subsets of sizes $\mathcal{M} = \{m_1, \dots, m_n\}$. The training objective minimizes the sum of reconstruction losses across all these levels, along with standard sparsity and auxiliary losses:

$$\mathcal{L} = \sum_{m \in \mathcal{M}} \mathcal{L}_{rec}(m) + \alpha \mathcal{L}_{sparse}, \tag{7}$$

where $\mathcal{L}_{rec}(m)$ is the reconstruction loss using only the first $m$ entries. This encourages the most important features to appear early in the dictionary, creating an ordered representation.

## D  LLM USAGE STATEMENT

We utilized a Large Language Model (LLM) to improve the grammar, spelling, and clarity of this manuscript. The authors critically reviewed and edited all suggestions and bear full responsibility for the accuracy and integrity of the final content.

| Prompt | Applied Attributes |
|---|---|
| Portrait of a woman in a flowing sundress in a field of wildflowers at golden hour | smiling, curly hair, laughing, old, surprised, young |
| Close-up of a woman in traditional Japanese kimono with cherry blossoms framing her face | smiling, curly hair, laughing, old, surprised, young |
| woman in business attire portrait in modern glass office building with city skyline | smiling, curly hair, laughing, old, surprised, young |
| Female pilot in leather jacket portrait next to vintage biplane | smiling, curly hair, laughing, old, surprised, young |
| woman in rain jacket portrait at lighthouse during coastal storm | smiling, curly hair, laughing, old, surprised, young |
| Portrait of a woman in bohemian clothing at outdoor art market in Paris | smiling, curly hair, laughing, old, surprised, young |
| Rock climber woman portrait with climbing gear and canyon background | smiling, curly hair, laughing, old, surprised, young |
| woman in winter coat portrait with Northern Lights in Finnish Lapland | smiling, curly hair, laughing, old, surprised, young |
| Female chef in whites portrait in busy restaurant kitchen | smiling, curly hair, laughing, old, surprised, young |
| Portrait of a woman in wetsuit on surfboard with ocean waves behind | smiling, curly hair, laughing, old, surprised, young |
| a portrait of a woman violinist in elegant gown in candlelit baroque chamber | smiling, curly hair, laughing, old,, surprised, young |
| woman in hiking gear portrait at mountain summit with valley vista | smiling, curly hair, laughing, old, surprised, young |
| Portrait of a man in a worn leather jacket with misty fjord background at dawn | smiling, curly hair, laughing, old, surprised, young, beard, bald |
| Portrait of a man in traditional samurai armor in a zen garden setting | smiling, curly hair, laughing, old, surprised, young, beard, bald |
| Portrait of a man wearing hiking gear with tropical canyon vista behind him | smiling, curly hair, laughing, old, smiling, surprised, young, beard, bald |
| man in fisherman's sweater portrait with foggy dock and sea background | smiling, curly hair, laughing, old, surprised, young, beard, bald |
| Young man in vintage band t-shirt leaning against 1967 Mustang in desert | smiling, curly hair, laughing, old, surprised, young, beard, bald |
| Portrait of a man in Renaissance clothing at an easel in Italian courtyard | smiling, curly hair, laughing, old, surprised, young, beard, bald |
| man in red flannel shirt portrait outside log cabin with falling snow | smiling, curly hair, laughing, old, surprised, young, beard, bald |
| male chef in whites at sushi counter, portrait with minimalist restaurant background | smiling, curly hair, laughing, old, surprised, young, beard, bald |
| man wearing panama hat portrait in Marrakech market with colorful spices | smiling, curly hair, laughing, old, surprised, young, beard, bald |

Table 3: The complete set of source prompts and their corresponding edit attributes used for our quantitative evaluation and user study.

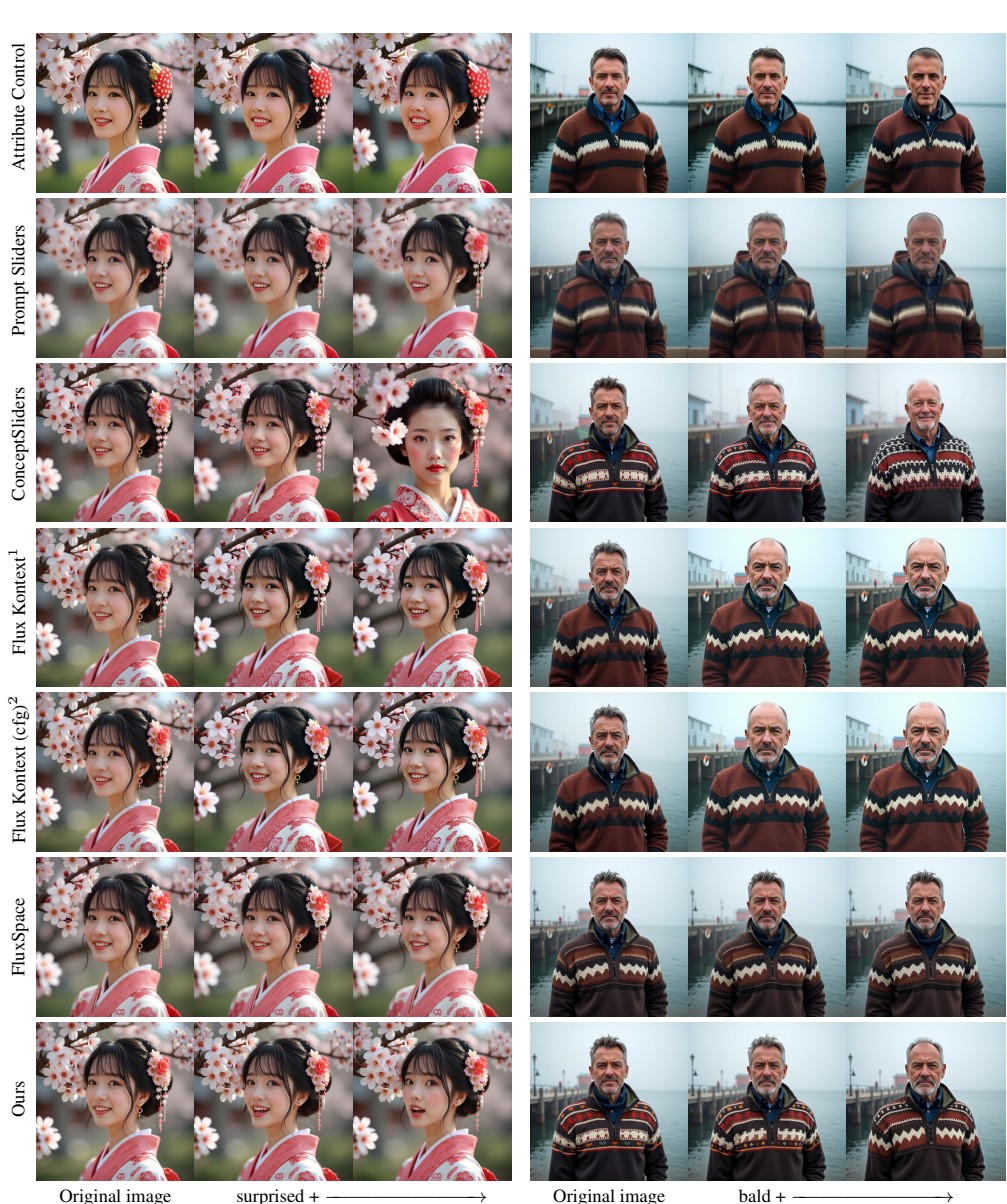

Figure 21: Each row showcases the results of a different editing method for the same edit. We now show two side-by-side runs (6 images per row). Our method (bottom row) produces a more disentangled result that better preserves the subject's identity compared to the competing approaches.

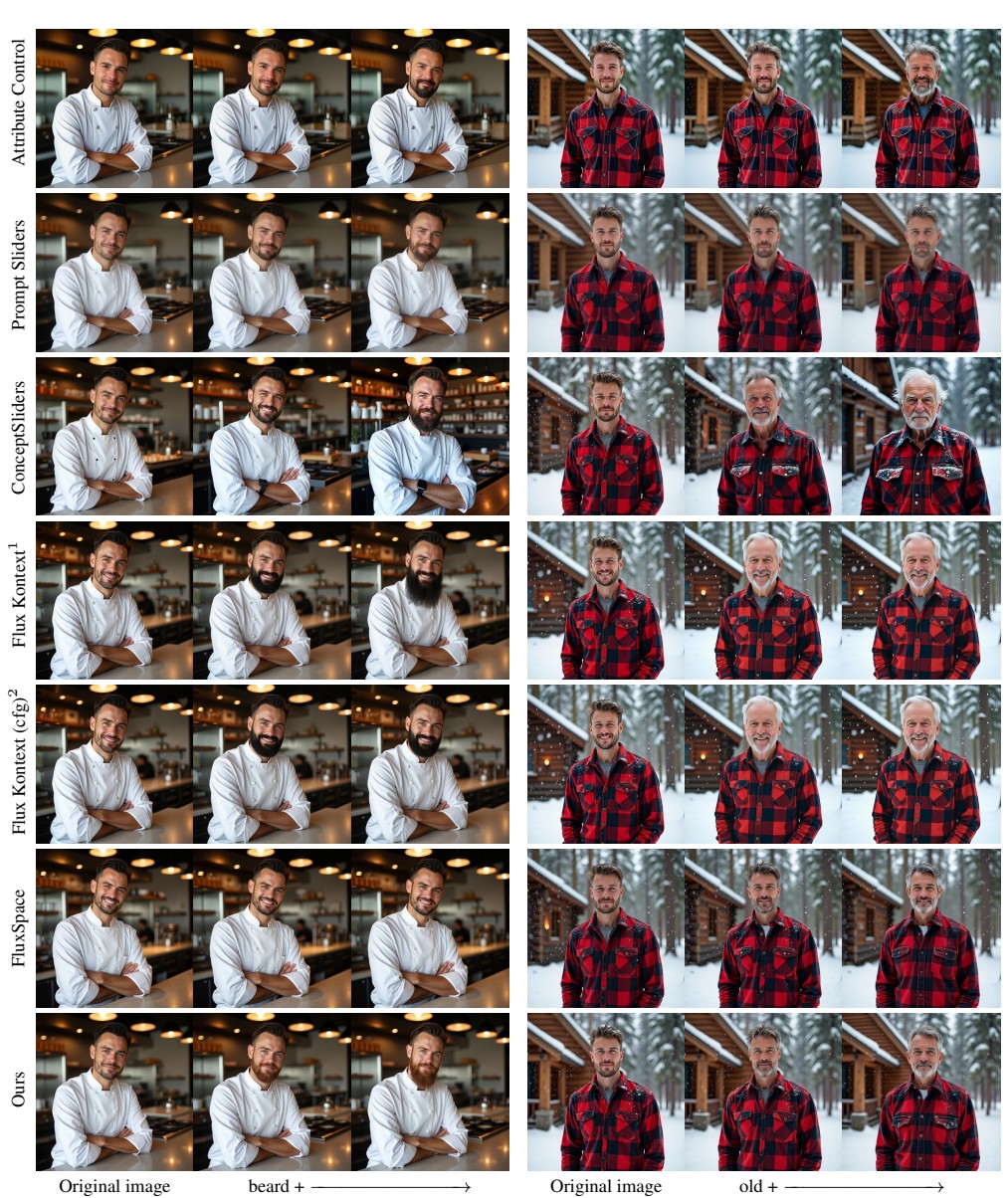

Figure 22: Each row showcases the results of a different editing method for the same edit. We now show two side-by-side runs (6 images per row). Our method (bottom row) produces a more disentangled result that better preserves the subject's identity compared to the competing approaches.

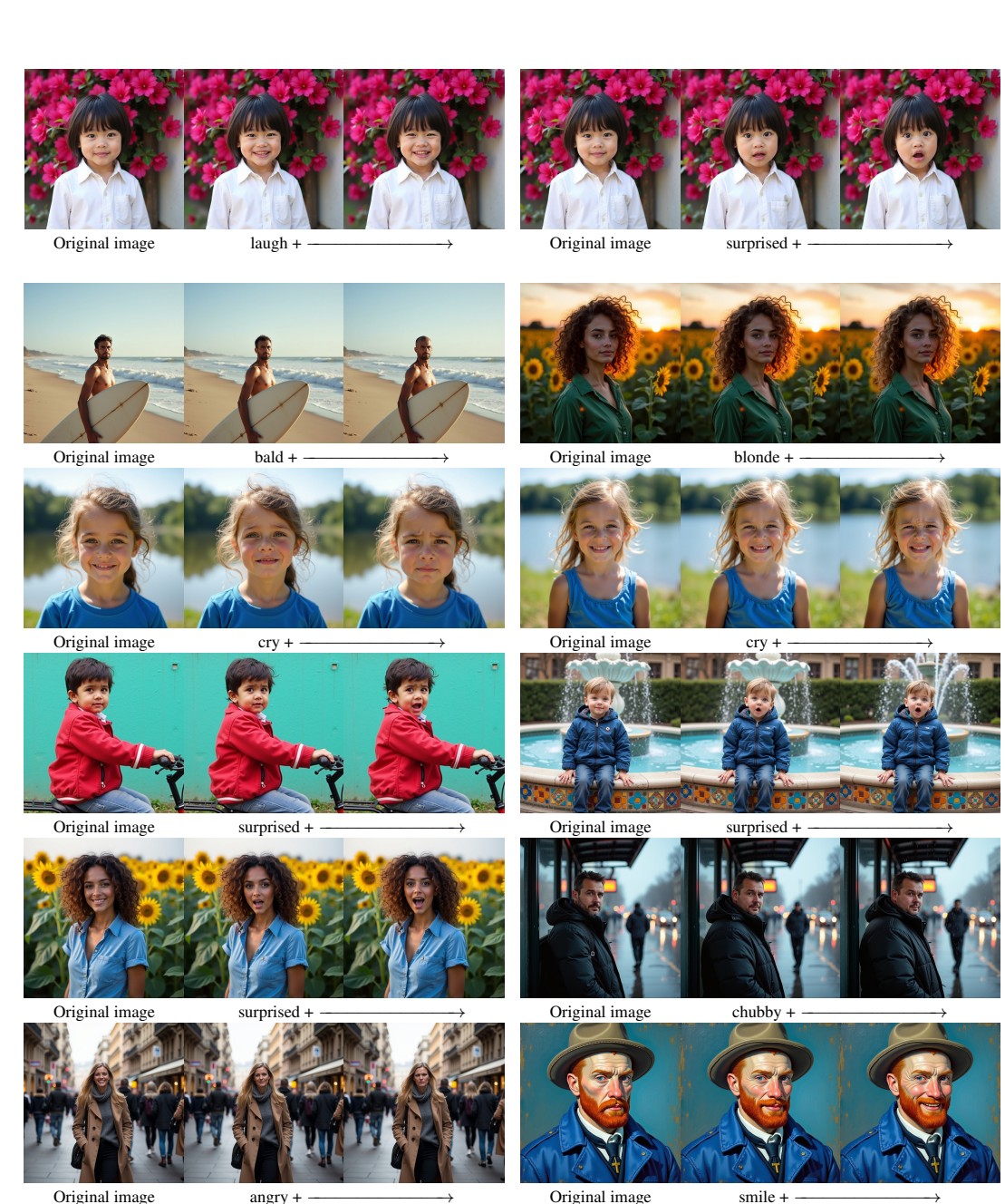

Figure 23: Additional results of our text-based Sliders.

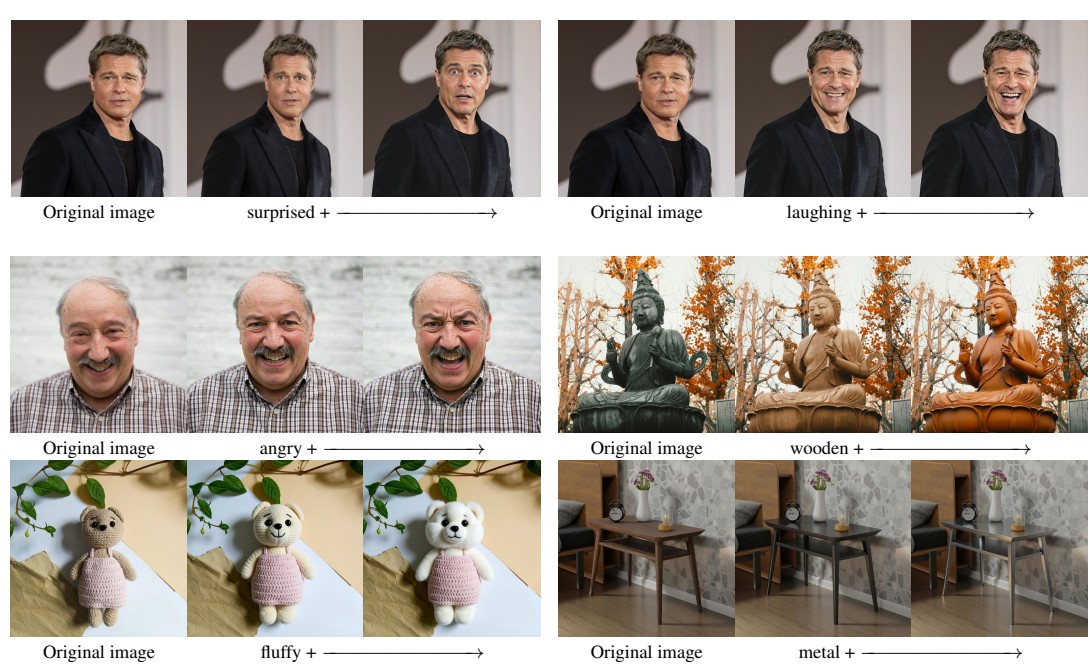

Figure 24: Additional results of our text-based Sliders employed on Flux-Kontext showcasing slider based manipulation on text to image instruction based models.

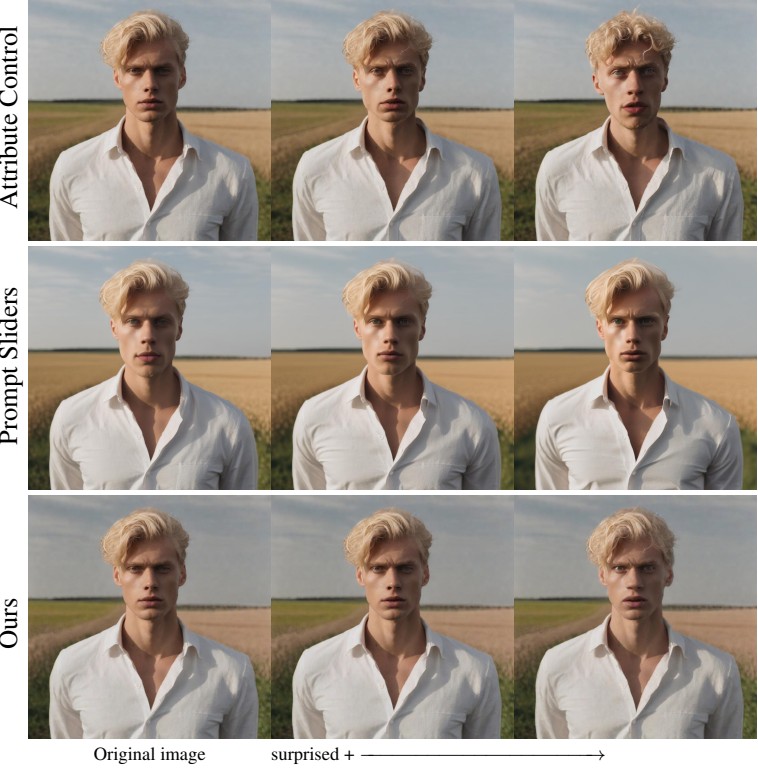

Figure 25: Comparing with SDXL-based methods. Since our method is Flux-based, we employ inversion on the SDXL-generated reference image when running our method to ensure a fair comparison. The SDXL baselines either fail to preserve identity (top) or produce a weak edit (2nd row). Our method (bottom) maintains the subject's identity while achieving a controllable attribute shift.

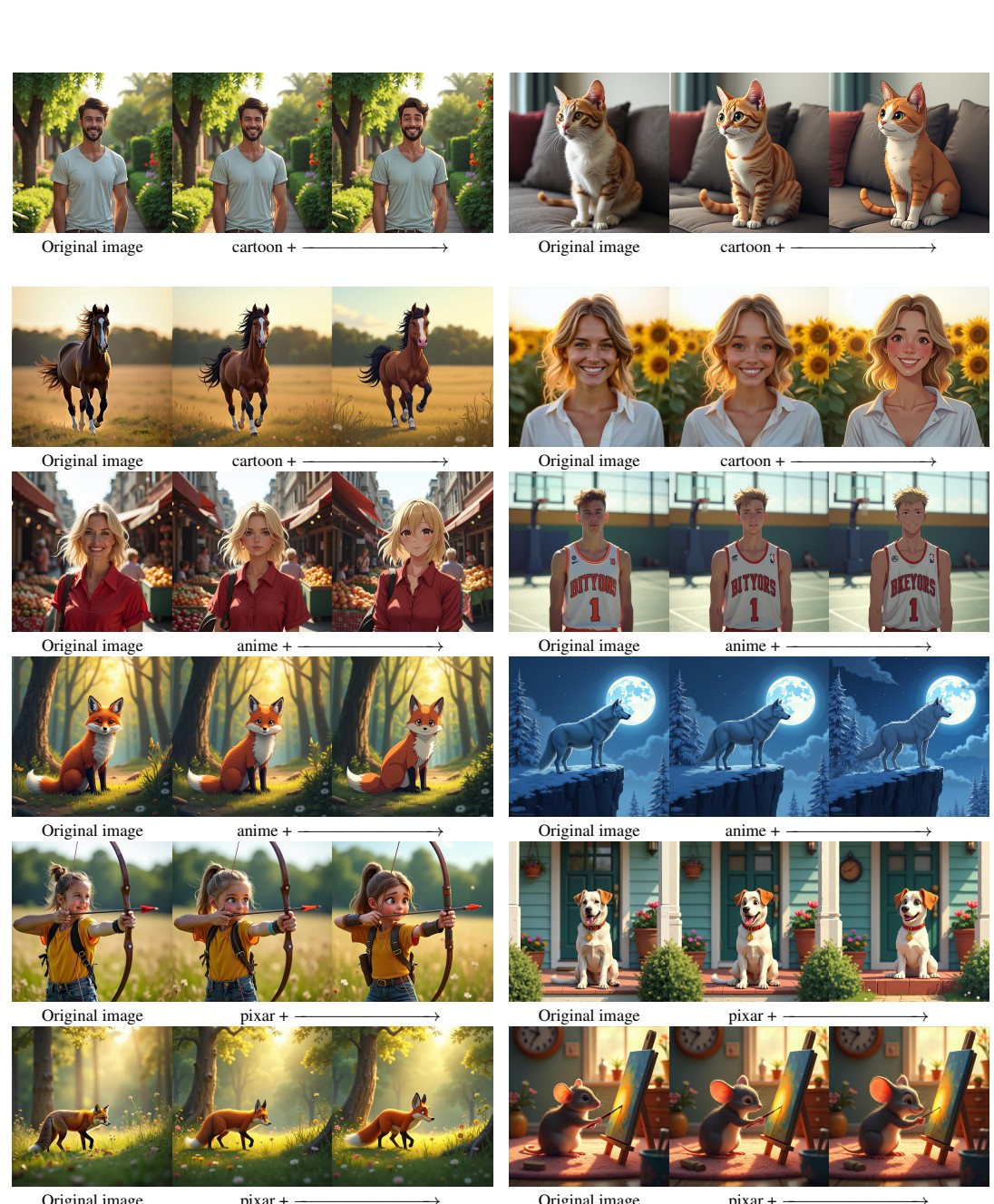

Figure 26: Additional results of our text-based sliders used for gradual style transfer

1512
1513
1514
1515
1516
1517
1518
1519
1520
1521
1522
1523
1524
1525
1526
1527
1528
1529
1530
1531
1532
1533
1534
1535
1536
1537
1538
1539
1540
1541
1542
1543
1544
1545
1546
1547
1548
1549
1550
1551
1552
1553
1554
1555
1556
1557
1558
1559
1560
1561
1562
1563
1564
1565

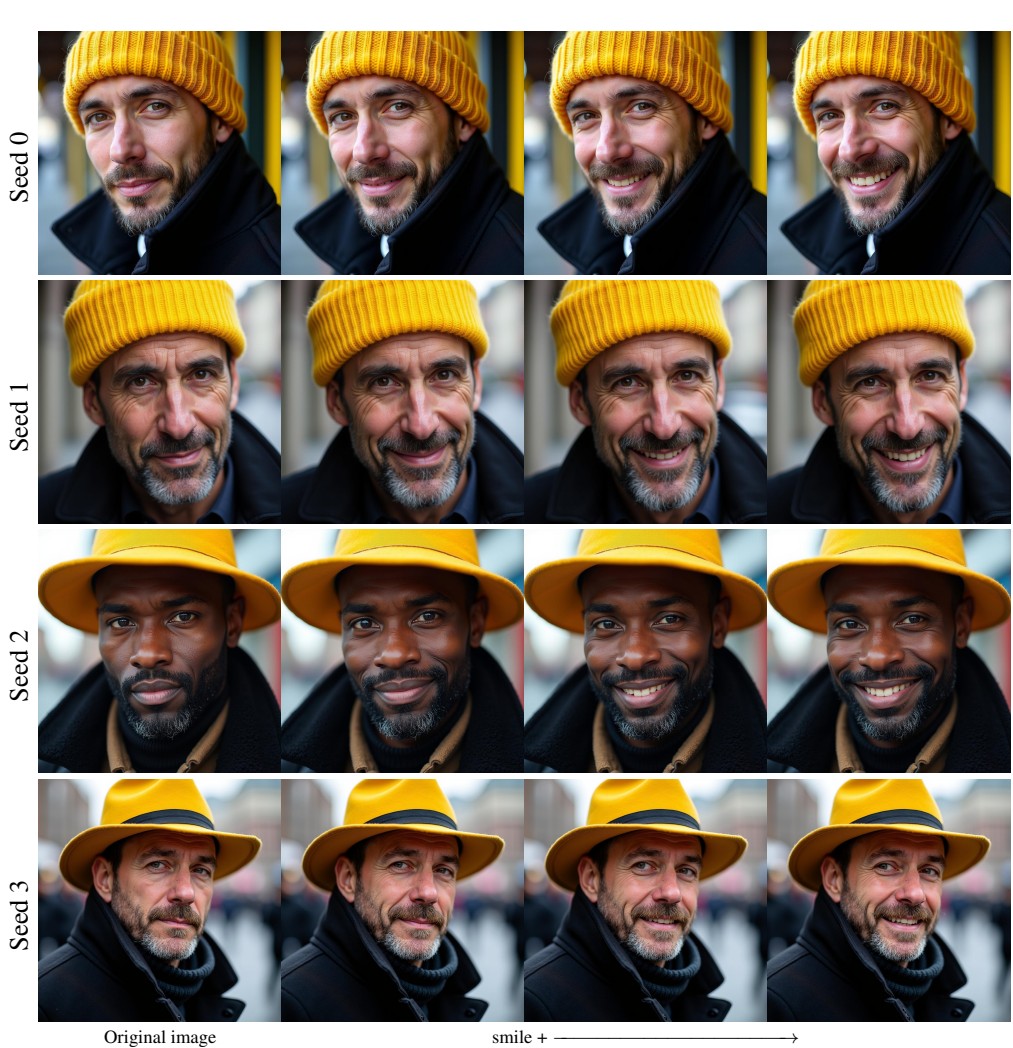

Figure 27: Consistency across Random Seeds. We apply the identical edit direction to images generated from four consecutive random seeds. This demonstrates the robustness of our method, which consistently applies the intended semantic attribute regardless of the initial noise or resulting scene variations.

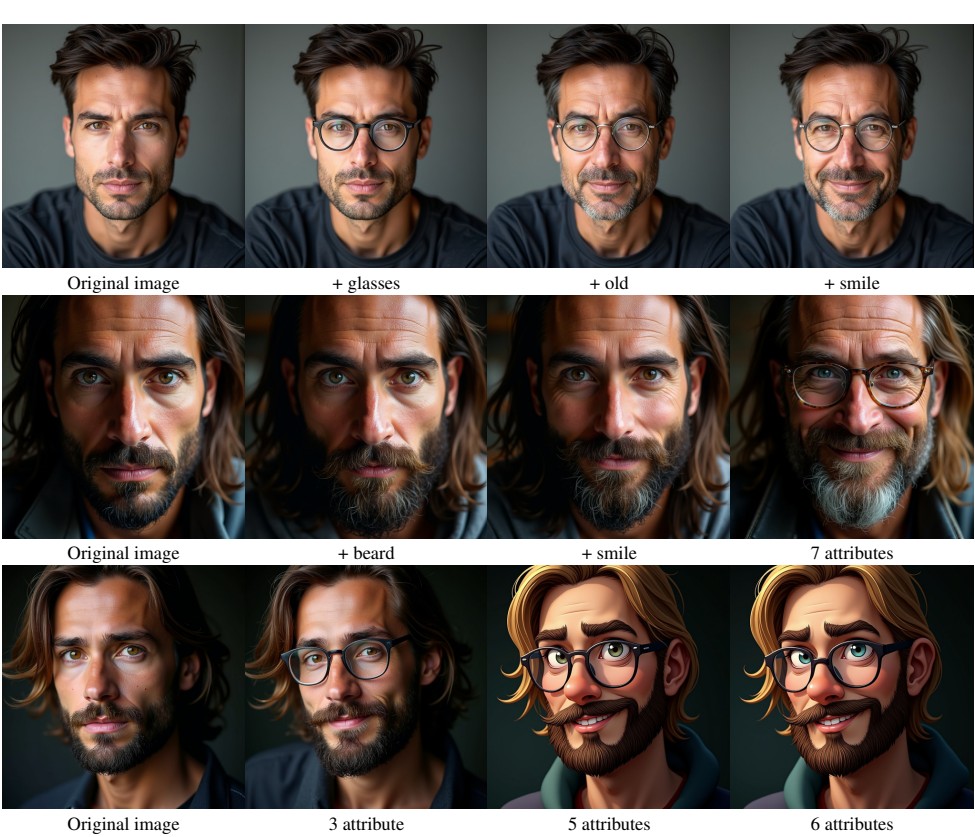

Figure 28: We show composition capabilities of our method. We apply an increasing number of sliders from left to right, showcasing our methods ability to concertante multiple edits on the same image. In the second row the added attributes are beard, smile, glasses, blonde hair, old, blue eyes, wide nose. In the third row the added attributes are glasses, smile, beard, cartoon, blonde, blue eyes.

