# OpenReview forum: "SAEdit: Token-level control for continuous image editing via Sparse AutoEncoder"
_ICLR.cc/2026/Conference — Submitted to ICLR 2026_

### Official Review · Reviewer_R5Rm · 2025-10-17

**Soundness:** 3
**Presentation:** 3
**Contribution:** 3
**Rating:** 6
**Confidence:** 3

**Summary:**

This paper introduces SAEdit, a novel method for achieving continuous and disentangled image editing by manipulating text embeddings at the token level. The core idea is to train a Sparse Autoencoder (SAE) on the output embeddings of a frozen T5 text encoder. This SAE maps the embeddings into a high-dimensional, sparse latent space where semantic attributes are isolated.

The primary contributions are: 1) A new framework that enables continuous and disentangled editing in diffusion models. 2) A model-agnostic approach that operates purely on text embeddings, making it compatible with various T5-based backbones (demonstrated on Flux and Stable Diffusion 3.5) without retraining. 3) Strong qualitative and quantitative results, evaluated via LPIPS, a VQA-based score, and a user study, which show it outperforms existing methods.

**Strengths:**

This paper introduces SAEdit, a novel and effective method for continuous and disentangled image editing by manipulating text embeddings. Strengths are:
1. Good Novelty: The central idea of applying an SAE to the text embedding space to discover disentangled semantic directions is highly innovative. Previously, GAN was much better in terms of such continuous editing ability. This paper introduces SAEdit, which enables diffusion models to have the same ability and can be applied to general methods.

2. Great Disentanglement and Control: The paper effectively addresses two major challenges in image editing: disentanglement and continuous control. The qualitative results clearly show that SAEdit can do continuous target edit without changing the objective ID.

3. Generality: A key advantage of SAEdit is its model-agnostic nature. By operating only on the text embeddings, it can be seamlessly applied to different T5-based text-to-image backbones without any model-specific retraining.

4. Well Written and Organized: The paper is well-written, clearly structured, and easy to follow. The methodology is explained in detail, and the figures are highly illustrative.

**Weaknesses:**

1. Because of the sparse learned space, the edit directions actually cannot be determined with simple words. So this would rely on LLM to generate multiple prompts, which can enrich the edit directions, make it clearer. But maybe that's why most image editing works would have a reference image. Wonder if authors can use a reference image to guide the edit directions. Image would provide much enriched direction information than text.

2. Potential Need for Manual Refinement: The limitations section notes that for some complex edits, "manually selecting or de-selecting a few specific entries in the sparse direction vector can yield even more disentangled results." This suggests that the fully automated direction-finding process is not always optimal and may require human intervention for the best performance, which slightly reduces the method's practical autonomy.

3. As the authors rightly point out, the method is constrained by the biases and semantic entanglements of the underlying text-to-image model. The failure cases on OOD edits are a clear example. While this is a shared limitation for many editing methods, it's a key boundary on the method's capabilities.

4. Evaluation Metrics are not that clear. I understand this paper's task is hard to quantify, so they only use the VQA score and LPIPS. The most desired case is that authors also introduce an evaluation benchmark to quantify the results.

**Questions:**

The main concern is the evaluation. If authors can introduce a more valid evaluation way or explain thoroughly, I will increase my score.

Also, there exists a little concern about using LLM to enrich the edit direction. But I think this one might hard to deal with, because of the sparsity nature of text, compared with image.

---

> ### Author Response · Authors · 2025-11-23
>
> We thank the reviewer for their time and valuable feedback. We address the reviewer’s comments below.
>
> **Potential Need for Manual Refinement.** All results in the paper and in the benchmark were produced with a single, fixed set of hyperparameters; no manual tuning was applied. Our method uses thresholds to determine how many neurons are interpolated in the sparse space. As with any such mechanism, there is a tradeoff: very strict thresholds can yield edits that are too weak, while very loose thresholds may introduce unintended changes. We selected values that strike a strong balance across attributes. We agree that refining the neuron selection strategy is a promising direction for future work and could further enhance disentanglement.
>
>
> **The method is constrained by the biases and semantic entanglements of the T2I model.** We agree that methods operating only on text embeddings inherit the semantic biases and entanglements of the underlying text to image model. These limitations, rooted in model expressivity and training data, are shared by many editing approaches and cannot be fully resolved without architectural or dataset changes.
> Nonetheless, controllable in-domain editing is highly valuable in practice, and our method performs strongly in this setting. Moreover, because SAEdit is plug and play and operates purely on text embeddings, it can immediately benefit from future models with improved semantic representations.
>
>
> **Evaluation Metrics are not that clear.** We acknowledge that evaluating semantic editing quality, especially for global and continuous edits, is challenging, which is why prior work often relies on qualitative results and user studies. In our paper, we report LPIPS for structural preservation and a VQA based score for semantic alignment, and we visualize the trade off between these metrics across all edits. Full benchmark details, including prompts and edit categories, are provided in Appendix B.2 and Table 3. We will also release the benchmark images. We have further added an identity similarity metric in Figure 20 of the revised version to improve benchmark fidelity.

---

> > ### Comment · Reviewer_R5Rm · 2025-11-24
> >
> > Thanks for the authors' rebuttal. It has solved my concerns.

---

### Official Review · Reviewer_LKJC · 2025-10-26

**Soundness:** 2
**Presentation:** 3
**Contribution:** 2
**Rating:** 4
**Confidence:** 5

**Summary:**

The paper proposes a Sparse AutoEncoder to learn sparse representations of attributes for disentangled edits and continuous control in diffusion models. It identifies a robust edit direction by selecting the largest singular vector of the matrix of N steering vectors using N different prompts. The method does not require task-specific optimization and only uses text embeddings for manipulating the attributes. Experiments are shown on SD3.5 and Flux diffusion models using LPIPS and VQA score metrics along with a user study.

**Strengths:**

The paper is clearly written and easy to read.

The proposed text based attribute control is scalable as it can be applied to all model architectures sharing the same text encoder.

It does not require task specific optimization.

**Weaknesses:**

Text-based attribute control is not new as prior works such as Attribute Control or PromptSliders have explored this idea. The proposed SAE based attribute control method is also not new since it is just applying to this task and so novelty is limited.

The  proposed method is not thoroughly evaluated against existing methods. Qualitative results comparing to Attribute-Control and Prompt Sliders are missing. Attribute Control can also applied for disentangled edits shown in Figure 14, 15 and 16.

Only two attributes are shown for composing edits. Does it work with multiple concepts together? The claim of disentanglement is not properly justified although the method is motivated by this property. Quantitative evaluation of composing multiple attributes should be conducted to justify as only few qualitative results are not sufficient to prove its robustness.

Some of the attribute edits are fundamentally limited by the semantic representation learned by the text encoder. For example, Figure 2 only changes the beard without changing the hair or face for being old.

Quantitative results on Face-ID is missing. Some of the edits sometimes changes the identity of the persons, e.g., Figure 6 blonde and smile examples. Figure 13 second row.

How sensitive is it to the seed of the diffusion model? Is the performance consistent across different seeds?


What about the concepts not well represented by the diffusion model? Unlike methods like concept sliders that can learn from images, this is a fundamental limitation of text based slider methods.

**Questions:**

See the weaknesses above.


Implementation details missing. How many concepts or edit attributes were extracted? There are only a few qualitative results shown for few concepts. Is the method robust to different attributes or styles?

How does the method work with CLIP text encoder?

---

> ### Author Response · Authors · 2025-11-23
>
> We thank the reviewer for his time and his constructive feedback and valuable suggestions. We will address each point made by the reviewer.
>
> **Qualitative results comparing to Attribute-Control and Prompt Sliders are missing.** We appreciate the suggestion. We focused our qualitative comparisons on methods that operate on the same base model (Flux.dev) to enable a fair assessment of edit quality independent of the generated image. In the revision, we have added qualitative comparisons to both methods. We also added a quantitative comparison to Prompt Sliders in Figures [10,20]
>
> **Quantitative evaluation of composing multiple attributes should be conducted.** This is a valuable suggestion. To our knowledge there is no broadly accepted metric for multi attribute composition; one prior work proposes a protocol, but it is not widely adopted. Creating a fair benchmark, implementing it, and rerunning all baselines would be substantial and may conflict with other requested experiments within the rebuttal window. If the reviewer considers a composition metric critical, we can attempt a targeted version during the rebuttal. Otherwise, we will plan it for a later revision.
>
> **Figure 2 only changes the beard without changing the hair or face for being old.** Thank you for the observation. On closer inspection, the edit in Figure 2 also affects the hair, neck, and broader facial region, but these changes are subtle and not easily visible at the current resolution. In the revision, we include panels with zoomed in crops to make the full extent of the aging edit clear.
>
> **Quantitative results on Face-ID are missing, some of the edits sometimes change the identity.** In our qualitative assessment, we did not observe a substantial identity shift in Figure 6 or in the second row of Figure 13; however, we fully agree that identity preservation should be quantified. In the revised version, we included a quantitative face-identity preservation analysis to complement the qualitative examples in Figure 20.
>
> **Sensitivity to seed selection.** Our method is not sensitive to the diffusion seed. Both the user study and the quantitative evaluations average over multiple seeds per prompt, and we observed no seed-dependent variability in performance. In the revision, Figure 27 illustrates consistent edits across different seeds.
>
> **learn from images.** Our method is not intended to learn editing directions directly from images, it is designed for text based semantic editing. We argue that the breadth of concepts expressible in natural language, combined with practical advantages such as fast inference, no per edit training, and no need to load or swap LoRAs, offers substantial benefits. Together with our strong user study and quantitative results, these factors show that our approach provides significant advantages in the text based editing setting.
>
> **How many concepts or edit attributes were extracted?** We present 23 unique editing directions in the paper. New directions are easy to add: extracting one requires about 100 sentence pairs, which can be easily generated automatically with an LLM. The method is robust across a wide range of attributes, as reflected in these 23 directions, and it performs consistently across diverse artistic styles. In the revision, we expand to 33 unique editing directions and include qualitative examples showcasing the additional edits.
>
> **How does the method work with CLIP text encoder?** This is a good question. We did not train our method with CLIP. We used T5, which is the text encoder most commonly adopted in current state-of-the-art diffusion models. Given the number of requested experiments and the timeframe window, we cannot add a CLIP experiment during the rebuttal. In principle, the method is encoder agnostic because it trains an SAE on the text encoder, so the same procedure applies to CLIP. If required, we can train and evaluate an SAE with a CLIP based model for the final version.

---

> > ### Comment · Reviewer_LKJC · 2025-11-27
> >
> > Thank you to the authors for the rebuttal. It addressed some of my concerns. But the concern on novelty and the disentanglement property is not fully justified.
> >
> > The advantage of the proposed method over Attribute Control for disentangled edits is not clear. As mentioned in the review, some qualitative comparisons with prior method for disentangled edits and additional quantitative comparison will strengthen the contribution.
> >
> > For composing multiple attributes, are there any qualitative examples showing that it can work for 3 or more concepts for the same image? Concept sliders show composition up to 50 slider concepts.
> >
> > Regarding text-only based semantic editing, it should be discussed as one of the limitations of the paper. The authors mentioned fast inference compared to prior methods, what is the overhead of this method during inference time?
> >
> > My question on CLIP encoder is to know whether there is any influence of the semantic structure of the text encoders on the SAEs learned and if it is robust to different text encoders. This is because the quality of the edits depends on the semantic organization within the text encoder. If it is robust, this will make the method much more general as it can apply to more class of image generation models.

---

> > > ### Author Response · Authors · 2025-12-03
> > >
> > > We thank again the reviewer for his comments and hope our final revision answers the remaining concerns.
> > >
> > > **Disentanglement and Compositionality.** Regarding the concern about composing multiple attributes, we have directly addressed this in the revision.
> > > We demonstrate our method's robust capacity for disentanglement and composition in **Figure 28**, which showcases the successful concatenation of **up to 7 different attributes on a single image.** This exceeds the reviewer's request for 3+ concepts and demonstrates compositionality comparable to state-of-the-art methods like Concept Sliders.
> > >
> > > **Comparison to Attribute Control.** We respectfully disagree that the advantage over Attribute Control is unclear. Our method demonstrates distinct advantages in both performance and efficiency:
> > >
> > > * *Quantitative & Qualitative Superiority:* As evidenced by the quantitative results in Figures 10 and 20, and the qualitative comparisons in Figures 12, 21, and 22, our method consistently outperforms Attribute Control in achieving disentangled edits.
> > >
> > > * *Training-Free Efficiency:* A critical advantage of our approach is that it doesn't require additional training per concept. Attribute Control relies on two variants: a training-free version (which performs significantly worse) and a training-based version which requires ~8 hours to train on new concepts. Our method exceeds the quality of training-based methods without the computational burden of retraining for new concepts.
> > >
> > > **Inference Overhead.** To address the question regarding overhead, we benchmarked the runtime for generating 10 images on Flux-dev, using 40 inference steps.
> > >
> > > * *Setup Time:* Given a new set of 100 sentence pairs, it takes approximately 7.46 seconds to compute the specific edit direction on a single A100 gpu. This is a one-time calculation per concept.
> > >
> > > * *Generation Time:* Once the edit direction is computed, there is zero overhead during inference. Generation time remains distinctively fast at ~19.5 seconds per generation on A100 gpu, identical to the out-of-the-box Flux-dev model.
> > >
> > > **Text-Only Semantic Editing.** Regarding the comment on text-only editing as a limitation:
> > > We view text-only semantic editing not as a limitation, but as the specific scope and design choice of this work. Our goal was to push the boundaries of what is possible through language-driven editing without relying on external masks or reference images, focusing on the ease of use for the end-user.
> > >
> > >  **CLIP Encoder Robustness.** We appreciate the insight regarding the semantic structure of text encoders.
> > > We agree that testing robustness across different text encoders is valuable for establishing generality. We commit to adding an analysis including an SAE trained on CLIP embeddings in the final version of the paper to demonstrate the method's applicability to a wider class of image generation models.

---

### Official Review · Reviewer_iZV6 · 2025-10-28

**Soundness:** 2
**Presentation:** 2
**Contribution:** 1
**Rating:** 2
**Confidence:** 5

**Summary:**

This paper employs sparse autoencoders on text embeddings to obtain disentangled semantic representations, so as to enable continuous attribute-specific image editing. Experiments show the effectiveness of the proposed method in providing disentangled and continuous image edits.

**Strengths:**

1.	The idea of employing sparse autoencoder to obtain disentangled semantic representation from text embedding is reasonable.
2.	The visual results of image editing in the empirical study are promising.

**Weaknesses:**

1.	Extracting semantic directions from text has been explored in [1] but is not discussed in this work. A detailed empirical comparison between the newly proposed method and [1] is lacking, which makes it difficult to judge the advancement of this work over [1].
2.	Several recent works on text-to-image editing (e.g., [2–4]) are missing from the related work. These studies demonstrate disentangled and continuous editing, so comparing with them is necessary to better show the proposed method’s advantage.
3.	The paper should also discuss related works on applying sparse autoencoders for disentangled representation learning [4]. In particular, several recent studies [5,6] have already employed sparse autoencoders in text-to-image models. Therefore, the proposed idea appears not to be novel, and the contribution of this work is marginal.

**Questions:**

1.	Is the relevant source token manually determined? If so, the approach seems too heuristic and labor-intensive, as it requires identifying the token for each given source prompt.
2.	I wonder whether the hyperparameter $\rho$ is sensitive. It would be helpful to include a sensitivity analysis of this hyperparameter.

---

> ### Author Response · Authors · 2025-11-23
>
> We thank the reviewer for the constructive feedback. Most concerns center on our discussion and comparison with prior work. Our evaluation compares against four methods, three of which have two variants, yielding seven baselines. This is comparable to or exceeds prior editing papers, which compare against three methods [1,2,5,9] or four methods [3,4,6,7,8]. We discuss each cited work below and have added several new comparisons in the revision. We believe these additions and clarifications address the concerns, and we hope they will improve the paper’s rating accordingly.
>
> **Comparisons with Sega [1].** We agree that SEGA is a relevant reference and appreciate the suggestion to address it explicitly. In the revised version, we have added a dedicated discussion in the Related Work. We are also preparing an empirical comparison with SEGA to strengthen the evaluation and plan to include it within the rebuttal period.
>
> **Missing related work [2,3,4].** These works are related but address different settings, so the methods are not directly comparable. LOCO / T-LOCO [2] performs localized edits using explicit input masks, which prevents global edits such as age. Classifier guided semantic optimization [3] requires training a classifier and manual parameter tuning per attribute, whereas our method finds an edit direction in seconds without per attribute training. NoiseCLR [4] extracts unconditional edit directions and is therefore not applicable to text based editing, since one cannot request a specific edit direction. The same authors later introduced FluxSpace, which is text based, and we already compare against it in our experiments. In the revision, we include a focused discussion of [2,3,4] in the Related Work.
>
> **Previous works about SAE in diffusion models [5-7].** While existing works focus on using the SAE for interpretability of diffusion models internals [6-7] (trained on the image’s diffusion features). The gap between these interpretability objectives and text based editing is substantial. Paper [5] examines SAE based interpretability of scientific document embeddings and is not related to text to image editing. In the revision, we include a discussion of [5-7] alongside our existing coverage of SAE work in diffusion models.
>
> **The paper should also discuss related works on applying sparse autoencoders for disentangled representation learning [4].** NoiseCLR doesn’t apply sparse autoencoders, did you intend to refer to a different paper?
>
> **Source token selection.** We select the source token corresponding to the subject the user intends to edit (for example, “man,” “table,” “sword”). This enables fine-grained, entity-specific control in complex prompts and is one of the strengths of our approach.
>
> **Rho hyperparameter.** The hyperparameter Rho controls how many sparse neurons are used to construct an edit direction. In our experiments, very small Rho reduces disentanglement and the edit can leak into unintended attributes, whereas very large Rho dilutes the signal and weakens the edit. A moderate Rho produces stable, high quality edits across all tested models. We will include a sensitivity analysis in the appendix in the final revision.
>
> \
> [1] Brack, M., Friedrich, F., Hintersdorf, D., Struppek, L., Schramowski, P., & Kersting, K. (2023). Sega: Instructing text-to-image models using semantic guidance. Advances in Neural Information Processing Systems, 36, 25365-25389.
>
> [2] Rohit Gandikota, Joanna Materzynska, Tingrui Zhou, Antonio Torralba, and David Bau (2023). Concept sliders: Lora adaptors for precise control in diffusion models.
>
> [3] Yusuf Dalva, Kavana Venkatesh, and Pinar Yanardag (2024). Fluxspace: Disentangled semantic editing in rectified flow transformers.
>
> [4] Inbar Huberman-Spiegelglas, Vladimir Kulikov, and Tomer Michaeli (2023). An edit friendly ddpm noise space: Inversion and manipulations.
>
> [5] Gaurav Parmar, Krishna Kumar Singh, Richard Zhang, Yijun Li, Jingwan Lu, and Jun-Yan Zhu (2023). Zero-shot image-to-image translation.
>
> [6] Stefan Andreas Baumann, Felix Krause, Michael Neumayr, Nick Stracke, Melvin Sevi, Vincent Tao Hu, and Bj¨orn Ommer (2025). Continuous, subject-specific attribute control in t2i models by identifying semantic directions.
>
> [7] Dalva, Y., & Yanardag, P. (2024). Noiseclr: A contrastive learning approach for unsupervised discovery of interpretable directions in diffusion models. In Proceedings of the IEEE/CVF conference on computer vision and pattern recognition (pp. 24209-24218).
>
> [8] Chen, S., Zhang, H., Guo, M., Lu, Y., Wang, P., & Qu, Q. (2024). Exploring low-dimensional subspace in diffusion models for controllable image editing. Advances in neural information processing systems, 37, 27340-27371.
>
> [9] Vladimir Kulikov, Matan Kleiner, Inbar Huberman-Spiegelglas, and Tomer Michaeli (2025). Flowedit: inversion-free text-based editing using pre-trained flow models.

---

> > ### Comment · Reviewer_iZV6 · 2025-11-26
> >
> > Thanks for the detailed responses.
> >
> > I would expect the authors to provide an empirical comparison with the relevant work SEGA. I also hope they can offer more detailed discussions of [5–7] in relation to this work to better position its contributions, as the current responses are too abstract.
> >
> > As for [4] referenced in W3, I am using it solely to illustrate disentangled representation learning techniques in T2I image editing.

---

> ### Author Response · Authors · 2025-12-03
>
> In our previous response we have added experiments and explanations to fulfill the reviewers request. We are now addressing the last requests submitted by the user,  by including a comparison to SEGA and expanding the related work section to discuss SAEs in text-to-image models.
>
> Specifically, we have added comparisons with SEGA to **Figure 10** and **Figure 20**, showcasing that our method has superior results. We also added a new discussion in Section 3 (highlighted in blue) that underscores our unique contributions compared to other SAE-based methods.
>
> To clarify the distinction between our work and prior literature:
>
> **Regarding [1, 2]:** In these works,  SAE’s are trained on the cross-attention blocks of the transformer to analyze the generation process of SDXL. Unlike our approach, [1, 2] are strictly analysis methods, not editing methods, and do not offer controllable text-based image editing. In contrast, we train SAEs on textual embeddings and introduce a sophisticated neuron selection mechanism. This allows us to identify interpretable editing directions for precise, gradual, and localized editing on specific tokens.
>
> **Regarding [3, 4]:** These works utilize SAEs for concept unlearning or steering towards safe generation. However, they do not provide mechanisms for fine-grained control. Our method uses a distinct neuron selection mechanism to identify disentangled directions, enabling the fine-grained control lacking in these prior works.
>
>
> [1] Surkov, V., Wendler, C., Mari, A., Terekhov, M., Deschenaux, J., West, R., Gulcehre, C., & Bau, D. (2025). One-Step is Enough: Sparse Autoencoders for Text-to-Image Diffusion Models. arXiv preprint arXiv:2410.22366. \
> [2] Surkov, V., Wendler, C., Terekhov, M., Deschenaux, J., West, R., & Gulcehre, C. (2025). Unpacking SDXL Turbo: Interpreting Text-to-Image Models with Sparse Autoencoders. arXiv / OpenReview preprint. \
> [3] Cywiński, B., & Deja, K. (2025). SAeUron: Interpretable Concept Unlearning in Diffusion Models with Sparse Autoencoders. arXiv preprint arXiv:2501.18052.\
> [4] Kim, D., & Ghadiyaram, D. (2025). Concept Steerers: Leveraging K-Sparse Autoencoders for Test-Time Controllable Generations. arXiv preprint arXiv:2501.19066

---

### Official Review · Reviewer_3R7L · 2025-11-01

**Soundness:** 3
**Presentation:** 3
**Contribution:** 3
**Rating:** 8
**Confidence:** 5

**Summary:**

The authors propose SAEdit, which aims to obtain disentangled and controllable editing directions to perform semantic edits on generated images. To achieve this task, the proposed method trains sparse autoencoders (SAEs) for source and target prompt pairs, using the embeddings obtained from the text encoder of the designated diffusion model (T5-XXL and FLUX, respectively). Over the provided qualitative and quantitative analyses, the method shows its effectiveness to be able to perform edits in a continuous way while preserving the disentanglement properties, with directions applied to the corresponding subject tokens.

**Strengths:**

- The method shows strong performance in terms of the continuity of the edits compared to the baselines. As summarized in Fig. 10, SAEdit achieves a continuous editing direction where the linearity of the effect can be quantified with the change in VQAScore.
- Editing directions trained with SAEdit do not require any extensive data collection for input-edit pairs, but operates with easily collectable prompt pairs.
- The method shows satisfactory performance over editing cases where the intended edit is ambiguous, which is challenging for the majority of the editing methods (e.g. surprised), as well as cross-domain.
- The quantitative experiments and user studies demonstrate the effectiveness of the method.

**Weaknesses:**

- The examples included in the paper are mainly human-centric examples, which may also be the case in the quantitative evaluations. The authors are strongly encouraged to provide more details on their evaluation setup and how diverse these 63 samples are.
- The examples for real image editing are limited, it should be clarified if the selective editing scheme is also valid for real image editing cases.
-  The limitations section can be extended with what are the types of edits that the method cannot tackle, if there are not such limitations, authors are encouraged to include more diverse examples for editing cases.

**Questions:**

- Does the proposed method require training of SAEs per generation prompt? From Fig. 5, it seems like the trained SAE is different for every generation prompt. If this is the case, what is the computational overhead that is required for every generation? For the robustness enhancement with SVD, with how much deviation training a new direction would be required?
- What are the generalization properties of the trained. SAE representations and their applicability to similar but slightly different prompts? As an example, if we train a SAE for the prompt pair ("a woman", "a smiling woman"), is this applicable to images generated with "a man" (to transform it into "a smiling man")?
- How does the method perform in stylization scenarios? Does the trained sliders handle gradual style changes?

---

> ### Author Response · Authors · 2025-11-23
>
> We thank the reviewer for his positive feedback and acknowledgment of our contribution, presentation and soundness. We address each point and the modifications we made in the revision to address it.
>
> **More details about the evaluation setup.** Full benchmark details, including all 63 prompts and the edited attributes, are provided in Appendix B.2 and Table 3. Upon acceptance we will release all the source images used for the benchmark.
>
> **Limited real image examples.** As shown in Figure 15, our method supports real image editing via inversion. In the revision, we also present another approach to slider based real image editing using FluxKontext, as described in Section 5.4.
>
> **Does the proposed method require training of SAEs per generation prompt?**  The SAE is pretrained once on a dataset of about 800 million tokens. It is not retrained per prompt, per edit, or per image. At inference, the only computation is a forward pass through the SAE encoder and decoder. If a new edit direction is computed, the robustness enhancement using SVD has negligible effect on image generation time compared with the rest of the diffusion model.
>
> **SAE generalization.** “if we train a SAE for the prompt pair ("a woman", "a smiling woman"), is this applicable to images generated with "a man" (to transform it into "a smiling man")?” - Yes. The learned direction generalizes across prompts and subjects. Because the SAE is trained on sentence pairs that describe the attribute shift, it captures the semantic difference between the neutral and target attributes, for example “smiling,” which enables transfer to other subjects such as “a man” or “a child.”
>
> **How does the method perform in stylization scenarios?** The method controls people’s attributes in stylized images, as shown in Figure 6. We also add qualitative examples of text based style sliders that gradually change an image’s style, as shown in Figure 26.

---

> ### Comment · Reviewer_3R7L · 2025-11-27
> **Thank you for the response**
>
> I appreciate the authors response to the raised questions and the supplementary examples provided. With the extended discussions for related works, examples and clarifications, I am keeping my rating and leaning towards the acceptance of the paper.

---

### Author Response · Authors · 2025-11-12

We thank all reviewers for their time and constructive feedback. We are currently addressing all comments and preparing a detailed response.

To reviewer iZV6: The references [1-6] seem not to be included. Could you please provide the full citations or titles so that we can address your comments?

---

> ### Comment · Reviewer_iZV6 · 2025-11-13
>
> Sorry for missing them. The references have been added.

---

### Author Response · Authors · 2025-11-23

We thank the reviewers for their thoughtful feedback and are encouraged that they found our method novel and effective (3R7L, R5Rm), with promising visual results (iZV6) and clear exposition (LKJC).

We have uploaded a revised manuscript that addresses the majority of the requests. Below is a summary of the key updates and additions included in this revision:

* **Expanded Edit Capabilities:** We have introduced 10 additional edit types to demonstrate the versatility of our method, specifically covering non-human subjects, real images, and style transfer. These results are visualized in Figures 17, 24, and 26.

* **New Quantitative Baselines:** We have updated Figure 10 to include a quantitative comparison against Prompt Sliders. In addition we added figure 20 that contains an ID consistency comparison between methods.

* **Additional Qualitative Comparisons:** To provide a more comprehensive evaluation, we have added visual comparisons against Prompt Sliders and Attribute Control in Figures 12, 21, 22, and 25.

* **Integration with Instruction Models:** We have added a new subsection (Section 5.4) detailing the integration of SAEdit with Flux-Kontext.

* **Improved Visualization:** We have added a zoomed-in crop to Figure 2 to enhance the visibility of the comparison.

* **Consistency Across Seeds:** We have added Figure 27 showcasing our method’s consistency across random seeds.

For the reviewers convenience all the altered parts in the revision are in blue.


*Common themes across reviews:*

**Human-centric focus.** We primarily evaluate human subjects for two reasons. First, facial editing is exceptionally challenging: human observers are highly sensitive to subtle attribute changes, making faces a stringent testbed for fine-grained, continuous control. Accordingly, many prior works focus on faces [1–7] in a variety of domains. \
Second, to ensure fair comparability with prior methods such as AttributeControl and ConceptSliders, we use their released weights and/or datasets, which are mostly human-centric (common attributes).
In the revision, we also report results beyond faces, including style control and other non-human categories, complementing the non-human examples already in the paper (Fig 8) and appendix (Fig 17).

**Novelty.**
We present a method that leverages an SAE trained on textual tokens for text-based image editing. This stands in clear contrast to prior SAE-based work (mentioned by reviewer iZV6 and discussed in our Preliminaries), which trains SAEs on diffusion activations for interpretability rather than editing.\
Unlike most editing approaches that intervene inside the generative model (e.g., manipulate attention maps) to obtain disentangled edits, we show that disentangled manipulations can be achieved entirely within the text embedding space, which serves as the model’s input. This is a surprising result given the discrete nature of textual prompts. It also makes the method model-agnostic, since operations in the shared text space naturally transfer across text-to-image models that use the same text encoder. \
To derive editing directions, we train an SAE on the T5 decoder’s token embeddings. The main difficulty is locating the neurons tied to the target edit. Because sparse activations are mostly zero and therefore do not overshadow one another, max-pooling in this space produces clear sentence-level concept embeddings, enabling our neuron-selection mechanism to highlight the key neurons.\
Together, these contributions enable continuous, disentangled, zero-shot attribute control without per-attribute training or test-time optimization.

**Quantitative evaluation**.
In line with prior work [8–12], we report LPIPS to quantify structural preservation. To further strengthen the analysis, the revision adds an ArcFace-based Face-ID similarity metric that measures identity preservation relative to the original subject.

For more specific questions or comments we address them directly to the relevant reviewer.

---

> ### Author Response · Authors · 2025-11-23
>
> [1] Qixun Wang, Xu Bai, Haofan Wang, Zekui Qin, and Anthony Chen (2024). InstantID: Zero-shot identity-preserving generation in seconds.
>
> [2] Guocheng Qian, Kuan-Chieh Wang, Or Patashnik, Negin Heravi, Daniil Ostashev, Sergey Tulyakov, Daniel Cohen-Or, and Kfir Aberman (2025). Omni-id: holistic identity representation designed for generative tasks.
>
> [3] Zheng Ding, Xuaner Zhang, Zhihao Xia, Lars Jebe, Zhuowen Tu, and Xiuming Zhang (2023). DiffusionRig: learning personalized priors for facial appearance editing.
>
> [4] Kaede Shiohara and Toshihiko Yamasaki (2024). Face2Diffusion for fast and editable face personalization.
>
> [5] Zhen Li, Mingdeng Cao, Xintao Wang, Zhongang Qi, Ming-Ming Cheng, and Ying Shan (2023). PhotoMaker: customizing realistic human photos via stacked ID embedding.
>
> [6] Haonan Lin, Mengmeng Wang, Yan Chen, Wenbin An, Yuzhe Yao, Guang Dai, Qianying Wang, Yong Liu, and Jingdong Wang (2024). DreamSalon: a staged diffusion framework for preserving identity-context in editable face generation.
>
> [7] Rishubh Parihar, Sachidanand VS, Sabariswaran Mani, Tejan Karmali, and R. Venkatesh Babu (2024). PreciseControl: enhancing text-to-image diffusion models with fine-grained attribute control.
>
> [8] Rohit Gandikota, Joanna Materzynska, Tingrui Zhou, Antonio Torralba, and David Bau (2023). Concept sliders: Lora adaptors for precise control in diffusion models.
>
> [9] Inbar Huberman-Spiegelglas, Vladimir Kulikov, and Tomer Michaeli (2023). An edit friendly ddpm noise space: Inversion and manipulations.
>
> [10] Gaurav Parmar, Krishna Kumar Singh, Richard Zhang, Yijun Li, Jingwan Lu, and Jun-Yan Zhu (2023). Zero-shot image-to-image translation.
>
> [11] Dalva, Y., & Yanardag, P. (2024). Noiseclr: A contrastive learning approach for unsupervised discovery of interpretable directions in diffusion models. In Proceedings of the IEEE/CVF conference on computer vision and pattern recognition (pp. 24209-24218).
>
> [12] Vladimir Kulikov, Matan Kleiner, Inbar Huberman-Spiegelglas, and Tomer Michaeli (2025). Flowedit: inversion-free text-based editing using pre-trained flow models.

---

### Comment · Area_Chair_kcGa · 2025-11-25

Dear Reviewers:

We kindly encourage you to take a moment to review the authors’ rebuttals and submit your feedback. Your prompt feedback is important for ensuring a thorough review. Thank you for your contributions to ICLR 2026. If you have responded to the authors' rebuttal, please feel free to ignore this message.

Thanks,
AC

---

### Author Response · Authors · 2025-12-03

We thank the reviewers for their constructive feedback. We are encouraged by the consensus regarding the core strengths of our work, specifically: **edit quality & continuity** (3R7L, iZV6, R5Rm), **generalizability** to models with shared text encoders (R5Rm, LKJC), **efficiency** (3R7L, LKJC, R5Rm), and **novelty** (R5Rm, 3R7L).

**In the Initial Revision** We addressed the broader concerns raised by the reviewers:
* *Scope:* Added 10 new non-human edit types (style/objects), real image examples, and Flux Kontext integration.
* *Novelty:* Clarified our direction-finding method (sentence embeddings/PCA) and distinct use of SAEs on textual embeddings.
* *Comparisons:* Added comparisons to Prompt Sliders and Attribute Control, plus a new Identity Preservation metric, demonstrating superior performance.

Addressing Remaining Concerns (Final Revision) **Reviewers R5Rm and 3R7L have confirmed their concerns are fully resolved**. We have now addressed the remaining points:

* Reviewer iZv6: We added an empirical comparison to SEGA (showing superior results) and expanded the Related Work section to cover additional SAE-based methods.
* Reviewer LKJC: We addressed the comparison to Attribute Control by highlighting our training-free advantage and superior quantitative benchmarks. To demonstrate disentanglement, we added a new figure (Figure 28) showing the successful concatenation of **7 distinct edit types** on a single image.

We believe the revision now comprehensively addresses all concerns and thank the AC and reviewers for helping us strengthen this paper.

---

### Meta-Review · Area_Chair_kT6G · 2026-01-05

**Summary:**

Across reviews, the main concerns consistently center on limited novelty, insufficient positioning with respect to prior work, and inadequate evaluation. The core ideas—semantic direction extraction, sparse autoencoder–based attribute control, and text-driven editing—have been explored in earlier studies, yet the paper lacks thorough qualitative and quantitative comparisons with representative methods (e.g., Attribute Control, PromptSliders, and prior SAE-based approaches), making the incremental contribution unclear. Moreover, the evaluation relies heavily on limited qualitative examples and a small set of metrics, without sufficient analyses of multi-attribute composition, identity preservation, robustness to random seeds, or hyperparameter sensitivity. Finally, reviewers note practical limitations such as reliance on manual intervention, LLM-assisted prompt expansion, and constraints imposed by the underlying text encoder, all of which require deeper discussion and empirical justification. Therefore, I recommend rejection.

**Reviewer Concerns:**

Reviewer R5Rm's concerns are addressed, Reviewer LKJC and Reviewer iZV6's concerns are still outstanding.

**Reviewer Scores:**

Reviewer LKJC may increase the score by 0–2 points.

---

### Decision · Program_Chairs · 2026-01-26

Reject